# Naphthoquinones Oxidize H_2_S to Polysulfides and Thiosulfate, Implications for Therapeutic Applications

**DOI:** 10.3390/ijms232113293

**Published:** 2022-10-31

**Authors:** Kenneth R. Olson, Kasey J. Clear, Paul J. Derry, Yan Gao, Zhilin Ma, Nathaniel M. Cieplik, Alyssa Fiume, Dominic J. Gaziano, Stephen M. Kasko, Kathleen Narloch, Cecilia L. Velander, Ifeyinwa Nwebube, Collin J. Pallissery, Ella Pfaff, Brian P. Villa, Thomas A. Kent, Gang Wu, Karl D. Straub

**Affiliations:** 1Indiana University School of Medicine—South Bend, South Bend, IN 46617, USA; 2Department of Biological Sciences, University of Notre Dame, Notre Dame, IN 46556, USA; 3Department of Chemistry and Biochemistry, Indiana University South Bend, South Bend, IN 46615, USA; 4Center for Genomics and Precision Medicine, Institute of Biosciences and Technology, Texas A&M Health Science Center, Houston, TX 77030, USA; 5Department of Chemistry, Rice University, Houston, TX 77005, USA; 6Stanley H. Appel Department of Neurology, Houston Methodist Hospital and Research Institute, 6560 Fannin Street, Houston, TX 77030, USA; 7Department of Internal Medicine, University of Texas—McGovern Medical School, Houston, TX 77030, USA; 8Central Arkansas Veteran’s Healthcare System, Little Rock, AR 72205, USA; 9Departments of Medicine and Biochemistry, University of Arkansas for Medical Sciences, Little Rock, AR 72202, USA

**Keywords:** reactive sulfur species, reactive oxygen species, antioxidants, juglone, plumbagin, vitamin K

## Abstract

1,4-Napththoquinones (NQs) are clinically relevant therapeutics that affect cell function through production of reactive oxygen species (ROS) and formation of adducts with regulatory protein thiols. Reactive sulfur species (RSS) are chemically and biologically similar to ROS and here we examine RSS production by NQ oxidation of hydrogen sulfide (H_2_S) using RSS-specific fluorophores, liquid chromatography-mass spectrometry, UV-Vis absorption spectrometry, oxygen-sensitive optodes, thiosulfate-specific nanoparticles, HPLC-monobromobimane derivatization, and ion chromatographic assays. We show that NQs, catalytically oxidize H_2_S to per- and polysulfides (H_2_S_n_, *n* = 2–6), thiosulfate, sulfite and sulfate in reactions that consume oxygen and are accelerated by superoxide dismutase (SOD) and inhibited by catalase. The approximate efficacy of NQs (in decreasing order) is, 1,4-NQ ≈ juglone ≈ plumbagin > 2-methoxy-1,4-NQ ≈ menadione >> phylloquinone ≈ anthraquinone ≈ menaquinone ≈ lawsone. We propose that the most probable reactions are an initial two-electron oxidation of H_2_S to S^0^ and reduction of NQ to NQH_2_. S^0^ may react with H_2_S or elongate H_2_S_n_ in variety of reactions. Reoxidation of NQH_2_ likely involves a semiquinone radical (NQ^·−^) intermediate via several mechanisms involving oxygen and comproportionation to produce NQ and superoxide. Dismutation of the latter forms hydrogen peroxide which then further oxidizes RSS to sulfoxides. These findings provide the chemical background for novel sulfur-based approaches to naphthoquinone-directed therapies.

## 1. Introduction

Quinones form the second largest class of antitumor agents approved in the U.S. for clinical use [1]. 1,4-Naphthoquinone (1,4-NQ), a quinone derived from naphthalene, is the most basic unit of the *para*-substituted naphthoquinones (NQs) and the structural parent of a variety of natural and synthetic NQs historically used in herbal medicine, and recently, of considerable clinical interest [2]. A number of these compounds, juglone, plumbagin, lawsone and vitamins K1 (phylloquinone), K2 (menaquinone) and K3 (menadione) are particularly noteworthy and have been extensively investigated. Juglone is commonly derived from walnut trees and plumbagin, from the roots of the medicinal herb *Plumbago zeylancia*. Lawsone (henna), obtained from the *Lawsonia inermis* tree, has been used since ancient times to dye clothing, hair and as a temporary tattoo [3]. The vitamin K compounds are well known as essential factors in blood coagulation and to treat osteoporosis [4]. Health benefits ascribed to these compounds include anti-inflammatory, anti-diabetic, antibacterial, antifungal, anti-atherosclerotic, analgesic, anti-osteoporotic, neuroprotective and hypolipidemic, but they are perhaps best known for their anticancer and antioxidant/prooxidant activities [5,6,7,8,9].

Collectively, NQs are well known as redox-cycling and alkylating agents, although much of their biological efficacy has been attributed to redox processes through the generation of reactive oxygen species (ROS), namely superoxide (O_2_^∙−^) and hydrogen peroxide (H_2_O_2_) [10,11]. These ROS may elicit cytoprotective effects in healthy cells by activating endogenous antioxidant responses or may kill ROS-sensitive tumor cells [10]. A number of studies have shown that NQs may be reduced by NADH or NADPH in reactions catalyzed by cytochrome P450 reductase or NAD(P)H:quinone oxidoreductase-1 (NQO-1), the former being an one-electron reduction to the semiquinone and the latter a two-electron reduction to the hydroquinone (reviewed in [11]). In either case, the reduced NQ is autoxidized by O_2_ yielding superoxide (O_2_^∙−^) which then dismutes, either spontaneously or catalyzed by superoxide dismutase (SOD), to O_2_ and H_2_O_2_. The latter (H_2_O_2_) accounts for most of the endogenous antioxidant response by activating the Keap-1/Nrf2 antioxidant system. Superoxide production, or at least its removal from the catalytic site, appears to be the rate-limiting step as the rate of redox cycling by NQs is enhanced in the presence of SOD [12]. Some of the confusion surrounding the biological functions of NQs is perhaps best illustrated by the mitochondrial-targeted naphthoquinone, SKQ1 [10-(6 -plastoquinonyl)decyl] triphenylphosphonium) which has recently been used as both a prooxidant [13] and a ROS scavenger [14].

We have recently shown that polyphenolic nutraceuticals in tea, berries and spices catalyze the oxidization of H_2_S to polysulfides and thiosulfate [15,16]. This reaction requires oxygen and appears to be mediated by initial autoxidation of a hydroquinone in the B ring of the polyphenol. We have also shown that a variety of quinones and hydroquinones, after the latter are autoxidized, oxidize H_2_S to polysulfides and thiosulfate in buffer and in cells [17]. Although side-chain modifications can augment catalytic efficacy, in general, *para*-quinones are more effective H_2_S catalysts than *ortho*-quinones, whereas *meta*-quinones are completely ineffective [17]. These findings suggest that 1,4-NQs may possess similar characteristics, and if so, a number of the physiological attributes of 1,4-NQs may be explained by their effects on cellular sulfur chemistry. In this study, we examine the effects of a variety of 1,4-NQs on inorganic sulfur metabolism in aqueous buffered solutions. We show that a number of these compounds catalytically oxidize H_2_S to thiosulfate and to inorganic per- and polyhydrosulfides, that this process consumes oxygen, and it involves redox cycling of the NQ. Chemical structures of the naphthoquinones used in this study are shown in Figure 1.

## 2. Results

### 2.1. Naphthoquinones Oxidize H_2_S to Polysulfides

We initially examined the effect of varying concentrations of 1,4-NQ on polysulfide formation (SSP4 fluorescence) from 300 μM H_2_S and observed that at low concentrations 1,4-NQ concentration-dependently increased SSP4 fluorescence, whereas above 10 μM, fluorescence progressively decreased; half as much fluorescence was observed with 30 μM 1,4-NQ compared to 10 μM (Figure 2A). Conversely, increasing concentrations of H_2_S continuously, and concentration-dependently increased SSP4 fluorescence when incubated with either 10 or 30 μM 1,4-NQ (Figure 2B). Juglone (Jug) and plumbagin (Pbn) also increased, then decreased polysulfide production with 300 μM H_2_S (Figure 2C,D). Maximum polysulfide production was observed at 1 μM for Jug and 10 μM for Pbn. Similar results were observed for 2-methoxy-1,4-naphthoquinone and phylloquinone, although there was no apparent inhibition at the highest NQ concentrations, anthraquinone and lawsone were minimally efficacious and menaquinone slightly decreased fluorescence compared to SSP4 with H_2_S (Appendix A). These results show that most NQs oxidize H_2_S to polysulfides and they suggest that the efficacy of this process is especially sensitive to the location, degree, and functional substitution (-OH, -CH3, -OCH3, etc.) substitutions on the 2, 3, and 5 positions of the NQ skeleton. The inhibitory effect of the NQs is examined in Section 2.2 below.

LC-ESI-HRMS was then used to further characterize the products of the H_2_S/1,4-NQ reaction. In this experiment, H_2_S (1 mM) was added to 1,4-NQ (1 mM) and allowed to react for 10 min. IAM (10 mM) was then added, and the sample was divided into three aliquots (Samples 1, 2, 3) which were serially analyzed at approximately 1 h intervals to monitor both the identity and stability of the IAM-polysulfides. As shown in Figure 2E, multiple inorganic polysulfides were produced after 10 min incubation of H_2_S with 1,4-NQ. The area under the curve (AUC) for H_2_S_2_, H_2_S_3_ and H_2_S_4_ increased between samples 1 and 2 but decreased for H_2_S_5_ and H_2_S_6_. The AUC for all samples decreased from sample 2 to sample 3. This suggests that either polysulfides, or IAM-derivatized polysulfides with *n* > 4, are somewhat unstable and may initially decompose to lower molecular weight polysulfides over the first hour and then later decompose to other sulfur moieties that are undetectable by LC-ESI-HRMS. Using values from sample 1, the percentage of the total polysulfide detected at 10 min was 39, 42, 15, 3 and 0.8% for S_2–6_, respectively, and for sample 2 was 57, 32, 8 and 1%. Collectively, this suggest that a variety of polysulfides are produced by 1,4-NQ catalyzed oxidation of H_2_S and that H_2_S_2_ and possibly H_2_S_3_ are likely the primary products of this reaction.

### 2.2. Reaction of 1,4-Naphthoquinone with Polysulfides Accounts for the Apparent Inhibition of SSP4 Fluorescence

Three experiments were designed to determine if the decrease in SSP4 fluorescence at NQ concentrations above 10 μM was due to some type of interference of the NQ with the SSP4 fluorescence, or due to decreased H_2_S oxidation by higher NQ concentrations. In the first experiment, 5 μM SSP4 was added to 10 μM of the mixed polysulfide (K_2_S_n_, *n* = 1–6) followed by 1,4-NQ to determine if 1,4-NQ interfered with the reaction between SSP4 and polysulfides. In the second experiment the order of SSP4 and 1,4-NQ addition was reversed to determine if the 1,4-NQ reacted directly with the polysulfide to produce a product undetectable by SSP4. In the third set of experiments K_2_S_n_ was incubated with SSP4 for 60 min to allow for the maximum SSP4 fluorescence to develop before adding 1,4-NQ to determine if the 1,4-NQ could reverse the reaction or if it inhibited fluorescence in other ways.

As shown in Appendix A, adding SSP4 to K_2_S_n_ before 1,4-NQ only slightly affected SSP4 fluorescence, whereas when 1,4-NQ was added first, it concentration-dependently decreased fluorescence to the extent that fluorescence was nearly completely inhibited by either 30 or 100 μM 1,4-NQ. SSP4 fluorescence was unaffected by 1,4-NQ when SSP4 was allowed to react with K_2_S_n_ for 60 min prior to addition of 1,4-NQ. Collectively, these results suggest that high concentrations of 1,4-NQ react with polysulfides to produce one or more products that are undetectable by SSP4.

### 2.3. Effects of Oxygen and SOD on Polysulfide Production Detected with SSP4

We have previously shown that low oxygen decreases SSP4-detected polysulfide production when H_2_S is incubated with CoQ_0_, whereas in normoxic solutions SOD increases polysulfides when H_2_S is incubated with CoQ_0_ [18]. This suggested that an examination of oxygen tension and SOD on 1,4-NQ reactions with H_2_S was also warranted. As shown in Figure 3, decreasing oxygen from 21% to <1% had minimal effects on SSP4 fluorescence at 110 min of incubation, whereas SOD increased fluorescence in both 21% and <1% O_2_. Overall polysulfide concentration at 110 min was greater with 10 μM than with 30 μM 1,4-NQ which is consistent with Figure 2 and the SOD-enhancement of SSP4 fluorescence is consistent with our previous observations on quinones [17,18]. However, the rate of increase in SSP4 fluorescence was considerably greater with SOD in <1% O_2_ than with SOD in 21% O_2_ with either 10 μM or 30 μM. It also appeared that there was a lag in the increase in SSP4 fluorescence over the initial 20–30 min with either 10 μM or 30 μM 1,4-NQ and SOD in 21% O_2_ that was not as evident in samples incubated at <1% O_2_. This suggests that oxidized 1,4-NQ (as provided by the supplier) is not reduced as quickly in 21% O_2_, as it is in <1% O_2_, an observation supported by absorbance spectra of these reactions (see, Section 2.6). We propose the delay in formation of reduced 1,4-NQ delays the establishment of a comproportionation reaction as discussed in Section 3.2.1.

We also noticed during our experiments that there was some day-to-day variation in the absolute amount of SSP4 fluorescence produced by incubation of NQs with H_2_S which we attribute to a number of possible factors including H_2_S volatility and differences in SSP4 lot numbers and/or the degree of SSP4 autoxidation over time. While this did not affect the clear trends of NQs to oxidize H_2_S, we felt a study of H_2_S oxidation by a number of NQs examined simultaneously, in both 21% O_2_ and <1% O_2_ was warranted. As shown in Appendix A, 1,4-NQ, plumbagin, juglone, 2-MNQ and the purported mitochondrial-targeted antioxidant, SKQ1 [14,19], concentration-dependently increased polysulfide production (SSP4 fluorescence) by oxidation of H_2_S. The apparent potency in 21% O_2_, in descending order, was plumbagin, 1,4-NQ = SKQ1, juglone, 2-MNQ. Consistent with previous experiments (described above), 30 μM NQ inhibited SSP4 fluorescence, except that this inhibition was not observed with 2-MNQ. The greatest increase in fluorescence was typically observed in the initial 20 min. Hypoxia (1% O_2_) typically decreased fluorescence, although it did not prevent it, and with juglone fluorescence produced by incubation of H_2_S with 10 and 30 μM NQs in <1% O_2_ was as great as maximal fluorescence at 21% O_2_. The effects of SOD on polysulfide production by other NQs were also examined. As shown in Figure 3E, SOD did not affect the small amount of polysulfide production by lawsone or juglone, whereas it greatly increased polysulfide production by 2-MNQ and menadione.

Collectively, these results confirm that a variety of NQs oxidize H_2_S to polysulfides, that slight modifications of the quinone affect overall efficacy of the reaction, and that decreasing oxygen decreases H_2_S oxidation but does not inhibit it, possibly because the NQs as obtained from the suppliers were mostly oxidized at the start of the reaction. The efficacy of SOD also appears to vary depending on modifications to the quinone ring.

### 2.4. 1,4-NQ Oxidizes H_2_S to Thiosulfate and Sulfite

#### 2.4.1. Thiosulfate Detection with AgNP

Thiosulfate is also a product of H_2_S oxidation by quinones [17,18]. Incubation of increasing concentrations of 1,4-NQ with 300 μM H_2_S exhibited a similar concentration-dependent increase in thiosulfate production from 1 to 100 μM 1,4-NQ and decrease in production at 300 μM (Figure 4A). Interestingly, the inhibitory effect on thiosulfate production with 300 μM 1,4-NQ was similar to the effect on polysulfide production, albeit the latter occurred at a lower concentration of 1,4-NQ.

Sparging the buffer with N_2_ and adding 1,4-NQ to H_2_S in a hypoxia chamber where the O_2_ was less than 1% somewhat decreased thiosulfate production compared to experiments simultaneously conducted in 21% O_2_ but did not inhibit it (Figure 4B). SOD increased thiosulfate production in both 21% and 1% O_2_. These results are consistent with the effects of O_2_ and SOD on polysulfide production measured with SSP4 (Figure 3).

The SOD-mediated increase in thiosulfate production could be due to shifting the reaction equilibrium by removing one of the products, namely superoxide, as proposed by Song and Buettner [12], or it could be due to H_2_O_2_ produced by dismutation of superoxide which would then oxidize H_2_S, although this reaction appears to be too slow to have a significant impact [20]. To examine these possibilities, thiosulfate production was measured in the presence of SOD and catalase (Cat, Figure 4C). With only H_2_S, the thiosulfate concentration was decreased by Cat and slightly increased by SOD + Cat suggesting that some of the background thiosulfate in the H_2_S stock was due to oxidation by oxygen. With H_2_S plus NQ, the thiosulfate concentration was greatly increased by SOD, decreased by Cat but unaffected by SOD + Cat. These results show that SOD augments thiosulfate formation, consistent with our previous observations on polysulfide formation. The results also show that catalase decreases thiosulfate concentration when incubated with H_2_S alone, or with H_2_S plus 1,4-NQ, and that Cat appears to completely inhibit the stimulatory effects of SOD when incubated with H_2_S and 1,4-NQ. This suggests that hydrogen peroxide contributes to much of the thiosulfate produced in these reactions, although it may not substantially contribute to polysulfude production from H_2_S [21].

#### 2.4.2. HPLC-MBB Detection of Thiosulfate and Sulfite

Thiosulfate and sulfite were also measured using HPLC separation of MBB-derivatized samples of the reaction between 300 μM H_2_S (as Na_2_S) and 10 μM 1,4-NQ. This produced 43 μM thiosulfate and 15 μM sulfite (average of duplicate samples). These results confirm that thiosulfate is produced by 1,4-NQ oxidation of H_2_S, and they also provide evidence for significant production of sulfite.

#### 2.4.3. Ion Chromatography (IC) Detection of Thiosulfate, Sulfite and Sulfate

To compensate for the relative insensitivity of the IC method, 3 mM H_2_S was added to 300 μM of either 1,4-NQ or 2-MNQ to evaluate production of thiosulfate, sulfite and sulfate. Both NQs significantly increased production of all three sulfoxides (Figure 4D). Upon subtracting the background sulfur compounds present or formed in the H_2_S blank during incubation, it was evident that thiosulfate was the predominant sulfur product followed by sulfite and then sulfate and that 2-MNQ tended to be somewhat less efficacious than 1,4-NQ in producing thiosulfate and sulfate. These results support the observations that thiosulfate was detected by both AgNP and HPLC-MBB methods and they add sulfate to the list of species produced by NQ oxidation of H_2_S.

### 2.5. H_2_S Reaction with Naphthoquinones Consumes O_2_ and Is Variously Affected by SOD and Cat

We have previously shown that *para*-quinones oxidize H_2_S to polysulfides and consume oxygen in the process [17,18]. Here, O_2_ consumption by NQs was measured under different conditions to determine if NQs exhibited similar characteristics, and if this was inhibited by higher concentrations of NQ. We also examined the effects of SOD and Cat on oxygen consumption by H_2_S and 1,4-NQ.

As an example, 1,4-NQ, Jug and Pbn concentration-dependently increased O_2_ consumption when added to a fixed concentration of 300 μM H_2_S (Figure 5A–C). Contrary to the apparent inhibition of polysulfide production at higher NQ concentrations there was no evidence of any inhibitory effects on O_2_ consumption at the high NQ concentrations, and all the O_2_ was consumed in many experiments. Plumbagin appeared to be the most efficacious at the higher NQ concentrations. Baseline oxygen concentration was not affected by 100 μM 1,4-NQ in the absence of H_2_S (Figure 5D). In a like manner, varying the concentration of H_2_S resulted in a concentration-dependent increase in O_2_ consumption in the presence of a fixed concentration of either 10 or 30 μM 1,4-NQ without any apparent inhibitory effects (Figure 5E,F). These experiments confirm that NQ-mediated oxidation of H_2_S is not inhibited by high concentrations of either 1,4-NQ or H_2_S.

It also became apparent that oxygen tension began to increase after incubating H_2_S with high concentrations of 1,4-NQ for an extended period. We attribute this to oxygen diffusion from the atmosphere into the buffer after one, or both of the reactants had been depleted. To determine if this was due to consumption of either H_2_S or 1,4-NQ, or both, we added a second doses of H_2_S or 1,4-NQ after oxygen tension began to increase while continuously monitoring oxygen levels. As shown in (Figure 5G,H), a second dose of H_2_S produced an additional decrease in oxygen, whereas a second treatment with 30 μM 1,4-NQ was ineffective. These results indicate that H_2_S is depleted in these reactions, but 1,4-NQ is not. This supports our hypothesis that 1,4-NQ functions as a catalyst. This hypothesis is further supported by the stoichiometry of the reaction, 30 μM of 1,4-NQ will consume more than 200 μM of oxygen and over 300 μM of H_2_S. H_2_S-1,4-NQ-mediated oxygen consumption was unaffected by H_2_O_2_ (Figure 5I), suggesting that hydrogen peroxide may not contribute to the formation of various sulfur species in these reactions. This is supported by previous observations that similar concentrations of H_2_O_2_ and H_2_S do not produce appreciable quantities of polysulfides [21] or thiosulfate (18.9 ± 1.5 versus 21.4 ± 0.8 uM, *n* = 4 for 100 μM H_2_S versus 100 μM H_2_O_2_ plus 100 μM H_2_S, Olson and Gao, unpublished). Superoxide dismutase increased O_2_ consumption by H_2_S/1,4-NQ, Cat alone did not affect it, but Cat somewhat diminished the effect of SOD (Figure 5J). Menaquinone and 2-methoxy-1,4-NQ also consumed O_2_ (Figure 5K,L), whereas anthraquinone (Figure 5M) only slightly increased O_2_ consumption, O_2_ consumption was not affected by either phylloquinone (Figure 5N) or lawsone (Figure 5O), which is consistent with the observations that these last three compounds do not produce significant concentrations of polysulfides. Addition of SOD to the H_2_S NQ reaction produced a further increase in oxygen consumption by 1,4-NQ, plumbagin and juglone, less of an increase with menadione and 2-MNQ and did not appear to affect oxygen consumption by lawsone (Appendix A).

Additional experiments were also performed to further examine whether the SOD-mediated increase in oxygen consumption in the reaction of 1,4-NQ and H_2_S was (a) due to removal of superoxide, as proposed by Song and Buettner [12], (b) due to reduction of oxidized NQ and establishment of a comproportionation equilibrium, or (c) due to oxygen incorporation into an unknown sulfoxide product of H_2_S. Addition of DTT to 1,4-NQ in the absence of H_2_S consumed more oxygen than addition of H_2_S to 1,4-NQ. SOD did not augment oxygen consumption by DTT and 1,4-NQ but, instead appeared to produce an initial transient decrease in the rate of oxygen consumption (Appendix A). SOD produced a similar transient decrease in oxygen consumption by DTT with 1,4-NQ and H_2_S, but it did not appear to affect overall consumption of oxygen (Appendix A). Conversely, DTT did not affect polysulfide production (SSP4 fluorescence) when incubated with H_2_S alone, or with H_2_S plus 1,4-NQ, whereas SOD decreased SSP4 fluorescence when added to H_2_S and 1,4-NQ (Appendix A). Addition of ascorbic acid to 1,4-NQ, either without or with H_2_S also increased oxygen consumption and these reactions were unaffected by SOD (Appendix A, respectively). As shown in Appendix A, oxygen consumption by H_2_S and 1,4-NQ was increased by tempol which is a general redox reagent against hydroxyl radicals, hydrogen peroxide and superoxide [22], but unaffected by mannitol, a weak scavenger of hydroxyl radicals and possibly other ROS [23].

To further examine the effect of a reductant on oxygen consumption by oxidized NQs, DTT was added to juglone, menadione, 2-MNQ and lawsone (Appendix A, respectively). DTT greatly increased oxygen consumption when added to juglone, whereas it produced only a modest increase when added to menadione and did not affect oxygen consumption when added to either 2-MNQ or lawsone. SOD slightly increased oxygen consumption when added to DTT with juglone or DTT with 2-MNQ, decreased oxygen consumption when added to DTT with menadione and had no effect on DTT with lawsone. Collectively, these results show that addition of reductants and SOD to NQs has effects on oxygen consumption that are specific for individual NQs and similar to those of H_2_S. They also suggest that autoxidation of NQs, not sulfoxide production is the primary source of oxygen consumption. These observations are generally consistent with the effect of the respective NQs on polysulfide production. The possible reactions involved are discussed in Section 3.4.2. NQ autoxidation.

### 2.6. Absorbance Spectra of H_2_S Reaction with 1,4-NQ Are Consistent with Redox Cycling

Absorbance spectra of oxidized and reduced 1,4-NQ, H_2_S, mixed polysulfides and 1,4-NQ plus H_2_S are shown in Figure 6A. Oxidized 1,4-NQ had absorbance peaks at 246 nm and 252 nm, a shoulder at 261 nm and a broad peak at ~342 nm. Reduction with H_2_ produced a new, strong peak at 208, a second peak at 240 and a broad peak that extended from ~314 to 320 nm. Reduction with H_2_S, followed by sparging with N_2_ to remove excess H_2_S, produced similar peaks at 208 nm and 240 nm, however the broad peak became even broader with a maximum absorbance around 288 nm. Na_2_S alone at pH 7 had peaks at 203 nm and 228 nm (Figure 6B), although the 203 peak may be a non-volatile impurity as it disappeared upon purification of sodium sulfide (not shown). Mixed polysulfides produced by incubation of elemental sulfur (S_8_) with Na_2_S produced peaks at approximately 296 nm and 375 nm (Figure 6C). The broad 288 nm peak observed after addition of H_2_S to 1,4-NQ (Figure 6A) may be due to a polysulfide or it could be due to formation of a sulfur-NQ adduct, which will be examined in future experiments.

Time-resolved spectra of the addition of 115 μM of H_2_S to 230 μM 1.4 NQ (1:2 H_2_S:1,4-NQ) in aerobic (21% O_2_) or anaerobic (<1% O_2_) buffer are shown in the left panels in Figure 6D,E, respectively, and the right panels of each show the spectrum after the H_2_S absorbance was subtracted. The latter gives a clearer evaluation of the effects of H_2_S on the 1,4-NQ spectrum. Time after addition of H_2_S is shown on the far right of Figure 6E. Addition of H_2_S progressively decreased absorbance of the 246 nm and 252 nm (oxidized 1,4-NQ) peaks over the initial 2–3 min, whereas the 208 nm (reduced 1,4-NQ) peak became more prominent up to 248 s then decreased but it did not disappear until the end of the experiment (1576 s). The shoulder at 264 nm became a prominent peak and the broad 342 nm peak became less evident as it merges with other components. When H_2_S was added to 1,4-NQ in low oxygen (<1% O_2_), the 246 nm and 252 nm peaks completely disappeared by 42 sec and the 209 nm (208 nm) peak continued to increase throughout the duration of the experiment. The 262–263 peak became prominent as it was in 21% O_2_, but now it was clear that the broad 342 nm peak was lost, and it was replaced by a broad peak at 321 nm. These results suggest that H_2_S reduced 1,4-NQ to a hydroquinone in the absence of oxygen, whereas in the presence of oxygen 1,4-NQ was initially reduced but then became partially reoxidized, resulting in an equilibrium between oxidized and reduced species. The 264 nm peak, which could be either a semiquinone or a sulfur adduct, also became more prominent. The cause of the gently sloping decrease in absorbance from 264 nm to 500 nm observed in 21% O_2_ but less evident at <1% O_2_ is unknown. It may be due to accumulation of polysulfides (cf. Figure 6C), formation of a semiquinone [24], turbidity due to colloidal elemental sulfur, or some combination of these three.

Further characterization of the H_2_S-1,4-NQ reactions are shown in Appendix A. Addition of 236 μM H_2_S to 115 μM 1,4-NQ (2:1 H_2_S:1,4-NQ) rapidly decreased the 246 nm and 252 nm peaks and produced a transient peak at 238 nm that became further blue-shifted to 228 nm by 428 s, after which the amplitude steadily decreased for the remainder of the experiment. The 208 nm peak remained prominent and only slightly decreased over the course of the experiment. A shoulder at 268 nm appeared late. The effects of H_2_S on the 246 nm and 252 nm peaks of oxidized 1,4-NQ became more evident after the H_2_S spectrum was subtracted from the H_2_S plus 14, NQ spectrum and they appeared to be essentially eliminated by 248 s. The peak at 268 nm also was more evident after the H_2_S spectrum was subtracted. By 62 s a slight shoulder appeared around 298 nm that persisted until it blended in at 350 s. Deconvolution of the H_2_S/1,4-NQ spectra showed a first component (*a*) indicative of the original oxidized 1,4-NQ, a second component (*b*) that is suggestive of the H_2_S spectrum with an additional small peak at 258 nm. The third component (*c*) has a peak at 207 nm, a shoulder at 252 nm and a prominent new peak at 262 nm. The 207 nm peak is suggestive of H_2_S, although if H_2_S is subtracted out then the 208 nm peak is compatible with reduced 1,4-NQ. The 252 nm peak represents the oxidized 1,4-NQ, the new 262 nm peak is somewhat similar to the late-developing peak observed in the H_2_S-subtracted spectrum (Appendix A, column D).

When the 1,4-NQ concentration was increased to 230 μM with the same 236 μM H_2_S (1:1 H_2_S:1,4-NQ), the 228 nm H_2_S peak rapidly disappeared, and after subtraction of the H_2_S spectrum, the 246/252 peaks due to oxidized 1,4-NQ persisted for most of the experiment but became progressively smaller, the *b* component of the deconvoluted spectrum became essentially identical to the *a* component (oxidized 1,4-NQ) and the 258 nm peak disappeared. The peak in the *c* component appeared as a flat peak from 252–265 nm, perhaps reflecting some of the remaining oxidized 1,4-NQ. Further increases in H_2_S, with constant 230 μM 1,4-NQ (2:1 and 4:1 H_2_S:1,4-NQ), produced progressive decreases in both the H_2_S and 1,4-NQ characteristics of the *a* and *b* components, respectively, the 264 nm peak now became predominant, and an additional peak appeared at 319 nm with another small shoulder at 350 nm. Rate constants for the three deconvoluted spectra are shown in Table 1. Changes in the amplitude of the 208 nm, 228 nm, and 252 nm peaks as a function of time are shown in Appendix A. As expected, the main H_2_S peak (228 nm) decreased the fastest with a low H_2_S:1,4-NQ ratio, whereas it was relatively unaffected at high ratios of H_2_S:1,4-NQ. The 252 nm peak also was relatively unaffected at high H_2_S:1,4-NQ, most likely because it was rapidly reduced at the onset of the experiment.

### 2.7. 1,4-NQ Oxidation of H_2_S Does Not Increase DCF Fluorescence

The SOD-mediated increase in H_2_S oxidation by 1,4-NQ suggests that this involves a one-electron oxidation of the NQ that would concomitantly reduce oxygen to superoxide as a rate-limiting process. The superoxide that is produced would then dismute, spontaneously or catalyzed by SOD, to hydrogen peroxide and an excess of ROS is generally assumed to be one of the products of NQ reactions. In addition, with a high NQ:H_2_S ratio it is also possible that NQ will reduce H_2_O_2_ to hydroxyl radicals in a metal-independent Fenton reaction [1] which will also be detected by the non-specific ROS fluorophore, dichlorofluorescein (DCF) [25]. To determine if any of these possibilities result from NQ/H_2_S/O_2_ redox cycling, various reactions were monitored with DCF over 90 min. As shown in Appendix A, DCF fluorescence was unaffected by addition of up to 30 μM of either 1,4-NQ, juglone, plumbagin, lawsone, phylloquinone, menaquinone or 2-methoxy naphthoquinone to 300 μM H_2_S, although all NQs slightly, but significantly, decreased fluorescence at the higher concentrations. Likewise, DCF fluorescence was essentially unaffected when the NQ:H_2_S ratio was increased stepwise from 1:1 to 300:1. These results suggest that either ROS are not produced in this reaction, which seems unlikely, or that if ROS are produced, they rapidly react with H_2_S, or more likely polysulfides, to produce sulfoxides, e.g., thiosulfate and sulfite, which was observed.

The reactions of NQ with hydrogen sulfide produce primarily superoxide anions and secondarily by dismutation of superoxide to make hydrogen peroxide. So, in the presence of SOD in living cells, hydrogen peroxide will be the predominate product, which along with molecular oxygen can react with either hydrogen sulfide [20,26,27] or polysulfides [28,29] to make sulfide, thiosulfate and finally sulfate (see Section 3). This also suggests that perhaps one of the biological effects of NQ-mediated metabolism of H_2_S is to rapidly quench ROS through diversion into relatively innocuous sulfoxides.

### 2.8. Free Iron Is Not Involved in 1,4-NQ Oxidation of H_2_S

Transition metals are often present as trace impurities in many reactions. These, especially iron, may catalyze autoxidation of naphthoquinones by complexing with NQ anions to overcome spin restriction and facilitate transfer of electrons to O_2_ [12]. To determine if iron contributed to autoxidation of NQs, 1,4-NQ was incubated with H_2_S in the presence of diethylenetriamine pentaacetic acid (DTPA) and polysulfide production monitored with SSP4 fluorescence. As shown in Appendix A, DTPA did not affect NQ-catalyzed polysulfide production and it can be concluded that iron does not contribute to 1,4-NQ autoxidation.

## 3. Discussion

### 3.1. Background

Tarumi et al., 2019 [30] evaluated the rate determining steps in the Takahax process whereby H_2_S is oxidized to elemental sulfur by 1,4-dihydroxynaphthalene-2-sulfonate (NQSH_2_). Forty-six possible reactions in the initial H_2_S oxidation/NQ reduction reaction were identified. The energetically most favorable reaction was a one electron oxidation of H_2_S, and reduction of NQ (Equation (2)) followed by a one electron oxidation of a second H_2_S and further reduction of the semiquinone to NQH_2_ (Equation (4)), with molecular sulfur produced by the combination of the thiyl radicals (Equation (6)). They also predicted that reoxygenation of NQH_2_ was accomplished by two, sequential one electron oxidations of the NQH_2_, the first producing superoxide and the second using superoxide as the electron acceptor to produce peroxide (Equations (13) and (15)). Furthermore, they predict that peroxide could substitute for oxygen in Equation (13) to produce the semiquinone and hydroxyl and these products would then react with each other to produce NQ and water. The rate limiting steps in the reactions were the initial oxidation of H_2_S (Equation (2)) and the initial oxidation of NQH_2_ (Equation (13)). Consistent with this one electron process, it has also been reported that in the presence of ascorbate and oxygen, 1,4-NQ catalyzes a one electron oxidation of ascorbate and the 1,4-NQ semiquinone is oxidized by oxygen producing superoxide and 1,4-NQ [31], although the authors suggest that this only involves redox cycling between the oxidized quinone and the semiquinone. Our studies clearly show that even in the presence of oxygen reactions with H_2_S fully reduce 1,4-NQ to 1,4-NQH_2_.

### 3.2. Possible Mechanisms of Per- and Polysulfide, Sulfite and Thiosulfate Production by 1,4-NQ Oxidation of H_2_S

#### 3.2.1. Per- and Polysulfide Production

Previous work in our lab has shown that many polyphenolic nutraceuticals readily oxidize H_2_S to polysulfides [15,16] and subsequent investigations demonstrated that this can be attributed to the redox active quinone [17,18]. Given the many biologically similar attributes between polyphenols and naphthoquinones, we felt it was important to examine sulfur metabolism by the latter to determine if this was a common theme in many of these nutraceuticals and related quinones and if this could readily explain their biological attributes. These studies also afforded us the opportunity to further investigate the mechanism(s) involved.

The absorption spectra shows that 1,4-NQ is oxidized when obtained from the supplier and that it is initially, and rapidly reduced by H_2_S, or more likely the hydrosulfide anion, HS^−^ (Figure 6). This could occur by a single two-electron reduction that produces the hydroquinone (NQH_2_) and sulfane sulfur (S^0^; Equation (1)), or by two sequential one-electron reductions of 1,4-NQ that initially produce semiquinone (NQ^·−^) and hydrosulfide radicals (Equation (2)); here indicated as HS^·^. (Note: the dianion, S^·−^, may be more appropriate as the pK_a_ appears to be near 4.0 [32]). A second reaction either between the semiquinone and hydrosulfide radicals could produce the hydroquinone and sulfane sulfur (Equation (3)), or a reaction between the semiquinone and a new hydrosulfide anion would produce a second thiyl radical (Equation (4)). The persulfide can be produced by the sulfane reacting with another hydrosulfide anion (Equation (5)), or the combination of two hydrosulfide radicals (Equation (6)). It should be noted that the pKa of persulfide is such that the hydropersulfide anion (HS_2_^−^) is more likely [33] and it has been suggested that the autoxidation of NQH_2_ likely proceeds in two steps starting with the reduced NQ dianion (NQ^2−^) [12] which is in equilibrium with the hydroquinone, the latter predominating in our experiments at neutral pH. Note also that the pKa for polysulfides becomes progressively lower as the number of sulfur atoms increases [33], but for simplicity they are represented below as the fully protonated species.
HS^−^ + NQ + H^+^ → S^0^ + NQH_2_(1)
HS^−^ + NQ → HS^∙^ + NQ^·−^(2)
HS^·^ + NQ^·−^ + H^+^ → S^0^ + NQH_2_(3)
HS^−^ + NQ^·−^ + 2H^+^ → HS^·^ + NQH_2_(4)
S^0^ + H_2_S → H_2_S_2_(5)
2HS^·^ → H_2_S_2_(6)

A hydrosulfide anion and hydrosulfide radical can also combine to form a hydropersulfide radical (Equation (7)),
HS^−^ + HS^·^ → HS_2_^·−^ + H^+^(7)
which can combine with a second hydrosulfide radical to produce a polysulfide (Equation (8));
HS_2_^·−^ + HS^·^ → H_2_S_3_(8)

Alternatively, the persulfide radical could react with molecular oxygen to produce superoxide and the disulfide (Equation (9)).
HS_2_^·−^ + O_2_ + H^+^ → H_2_S_2_ + O_2_^·−^(9)

Polysulfides (S > 2) can also be produced by multiple reactions between sulfane and a persulfide (Equation (10)),
nS^0^ + H_2_S_2_ → H_2_S_(n+2)_(10)
or dimerization of persulfide radicals (Equation (11)),
2HS_2_^·−^ → H_2_S_4_(11)

Persulfide radicals may also be produced by reactions of polysulfides with the quinones [34] (Equation (12)),
H_2_S_2_ + NQ → HS_2_^·−^ + NQ^·−^ + H^+^(12)
two of which could then form a tetrasulfide or one plus a thiyl radical would form a trisulfide.

Reoxidation of NQH_2_ could occur via sequential one electron steps with oxygen as the electron acceptor (Equations (13) and (14))
NQH_2_ + O_2_ → NQ^·−^ + O_2_^·−^ + 2H^+^(13)
NQ^·−^ + O_2_ → NQ + O_2_^·−^(14)
or superoxide as the electron acceptor in the second reaction (Equation (15)),
NQ^·−^ + O_2_^·−^ + 2H^+^ → NQ + H_2_O_2_(15)

Alternatively, this could occur through a single two electron reaction (Equation (16)),
NQH_2_ + O_2_ → NQ + H_2_O_2_(16)
although this reaction is spin forbidden and should be extremely slow. Finally, the formation of the semiquinone intermediate can also occur by comproportionation of the quinone and the hydroquinone (Equation (17)),
NQ + NQH_2_ → 2NQ^·−^ + 2H^+^(17)

This reaction would serve to bypass the initial oxidation of NQH_2_ by molecular oxygen (Equation (13)) and thus overcome unfavorable energetics of the first oxidation step. Such a process is supported by our studies of the effects of SOD on SSP4 fluorescence and absorbance spectra in 21% and <1% O_2_ (Section 2.3 and Section 2.6, respectively). A number of these potential reactions are depicted schematically in Figure 7A–C.

#### 3.2.2. Sulfite and Thiosulfate Production

We have observed that, in addition to polysulfides, a variety of quinones oxidize H_2_S to thiosulfate, sulfite and sulfate [15,16,18]. Although we did not perform an extensive analysis of all possible sulfoxides that could be produced by the NQ/H_2_S reaction, we show that sulfite, sulfate and thiosulfate are produced.

There are numerous pathways for production of sulfoxides but the details of separate steps in the reactions are not entirely clear [35]. For example, Kleinjan et al. [28] studied the oxidation of polysulfides by oxygen and found production of thiosulfate which they suggest proceeds through an “nascent” elemental sulfur form at pH values less than 9. Steudel et al. [29] described the formation of thiosulfate and elemental sulfur from oxidation by molecular oxygen of polysulfides at pH > 9. At these elevated pH values, they did not see sulfide, sulfate or polythionates.

According to the studies done by Hoffmann [20], H_2_S is oxidized by hydrogen peroxide to make sulfite and sulfite is oxidized by peroxide to make sulfate. In our experiments superoxide is produced from oxidation of the reduced quinones and semiquinones. The well-known spontaneous dismutation of superoxide into hydrogen peroxide and oxygen will result in increasing concentrations of hydrogen peroxide which will enable the above reactions to proceed to make sulfite. Further reactions with hydrogen peroxide will lead to sulfate (Figure 4D).

Thiosulfate can arise from the reaction of elemental sulfur, derived from the two-electron reduction of quinones by hydrogen sulfide, which can then react with sulfite derived from the above reactions (Equation (18)),
S + SO_3_^2−^ → S_2_O_3_^2−^(18)

Sulfenic acid may be an important intermediate and can be formed from two possible reactions in our experiments (Equations (19) and (20)),
H_2_S + H_2_O_2_ → HSOH + H_2_O(19)
S + H_2_O → HSOH(20)

The sulfenic acid can react with hydrogen peroxide to give sulfoxylic acid (Equation (21)),
HSOH + H_2_O_2_ → HOSO^−^ + H_2_O + H^+^(21)
followed by a second reaction with sulfenic acid to make thiosulfoxylic acid (Equation (22)),
HSOH + HOSO^−^ → HOS_2_O^−^ + H_2_O(22)

Thiosulfoxylic acid can react with hydrogen peroxide to make thiosulfate (Equation (23)),
HOS_2_O^−^ + H_2_O_2_ → S_2_O_3_^2−^ + H_2_O + H^+^(23)

In addition, under alkaline conditions sulfoxylic acid can react with either superoxide or hydrogen peroxide to give the sulfur dioxide anion radical which can react with itself reversibly to give dithionite [35].

#### 3.2.3. Proposed Mechanism of 1,4-NQ Oxidization of H_2_S to Produce Polysulfides and Elemental Sulfur

Our experimental evidence suggests that the primary reactions are somewhat different from those offered by Tarumi et al. [30]. We propose that the initial reaction is a one-step, two-electron reduction of 1,4-NQ and concomitant oxidation of H_2_S to sulfane sulfur (S^0^) as shown in Equation (1). One-electron reduction potentials for 1,4-NQ [36] are −140 mV (NQ/NQ^·−^) and +306 mV (NQ^·−^/NQH_2_), whereas the one-electron reduction potential for HS^−^/HS +920 mV [37], making either one-electron reaction unlikely (see also Table 2). Furthermore, our spectral analysis shows that 1,4-NQ is rapidly and fully reduced to the hydroquinone (Figure 6 and Appendix A) and this would be difficult to achieve from a semiquinone intermediate. This suggests that the NQ does not redox cycle between the oxidized NQ and the semiquinone to oxidize H_2_S. Conversely, the two-electron reduction potential for 1,4-NQ is +83 mV [36] and the two-electron reduction potential for sulfur/hydrosulfide (S^0^/HS^−^) is −170 millivolts [38], both of which are quite favorable. Furthermore, S^0^ would readily react with H_2_S_n_ (where *n* = 1–5) to produce the variety of polysulfides we observed with mass spectrometry (Figure 2).

#### 3.2.4. Autoxidation of 1,4-NQH_2_

We also suggest that reoxygenation of fully reduced 1,4-NQH_2_ is different from that proposed by Tarumi et al., 2019 [30]. Once 1,4-NQ is reduced by H_2_S (HS^−^) it must be reoxidized, presumably by oxygen. A two-electron oxidation of the reduced naphthoquinones by molecular oxygen is spin-forbidden [41] suggesting that reoxidation proceeds via one electron oxidation steps and goes through semiquinones as intermediates. This process could proceed by either of two steps (or both). The first involves production of the semiquinone by oxidation of the reduced quinone by molecular oxygen to yield superoxide and the semiquinone, in the second, the semiquinone is produced by comproportionation of an oxidized and reduced naphthoquinone. The one-electron reduction potentials for NQ/NQ^·−^ and NQ^·−^/NQH_2_ are −140 mV and +306 mV, respectively, [12,41], so it would initially appear that is less likely that a substantial fraction of 1,4-NQH_2_ is initially oxidized by oxygen as only the second one-electron reduction mid-point potential for 1,4-NQ, is favored for autoxidation. However, although, reaction between 1,4-NQH_2_ with O_2_ as a one-electron oxidation of QH_2_, i.e., NQH_2_ +e → NQ^·−^ + O_2_^·−^ + 2H^+^ is also spin restricted [12,41], Song and Beutner [12] show that SOD increases autoxidation of 1,4-NQ. They [12] propose that this reaction (Equation (13)) can occur if the superoxide product is rapidly removed. Our observations that SOD increases polysulfide production (Figure 3) and oxygen consumption (Appendix A) supports this argument. However, we propose that the oxidation of 1,4-NQ to a semiquinone is also accomplished by comproportionation of 1,4-NQ and 1,4-NQH_2_ the former being present in the initial stock from the supplier, and the latter having been formed by two-electron reduction of 1,4-NQ by H_2_S. Either way, once the semiquinone is produced it then reacts with molecular oxygen to produce 1,4-NQ and superoxide. Further discussion of the autoxidation processes of 1,4-NQ and other NQs used in this study are described in Section 3.4.

The loss of the 246/252 nm absorbance peak when 1,4-NQ was exposed to H_2_S in hypoxia (Figure 6E) indicates that 1,4-NQ is completely (or nearly completely) reduced to 1,4-NQH_2_ relatively quickly in the absence of another oxidant. That the oxidized 1,4-NQ peak only slowly decreased when H_2_S was added to 1,4-NQ in 21% oxygen is also indicative of redox cycling of the oxidized and reduced NQ. It is known that the semiquinone of benzoquinone has a strong absorption in the 400–500 nm range when the semiquinone is unionized and that this absorbance shifts to the 200–300 range for the ionized species at pH > 4 [24]. If the same effects hold for naphthoquinones a semiquinone adduct should also appear in the 200–300 nm range, which may be the peak we observed.

### 3.3. Effects of Superoxide Dismutase and Catalase

As described in the introduction, much of the biological effects of NQs are ascribed to their production of superoxide and hydrogen peroxide through reduction of molecular oxygen. We and others have observed that SOD increases redox cycling of quinones, which suggests that production of superoxide is a rate limiting step in overall reaction [12,18]. This also appears to be the case for NQ (Figure 3 and Figure 4, Appendix A).

Regarding per- and polysulfide production, our mass spectrometry analysis (Figure 2E) suggests that persulfide production predominates in the initial reaction and that either longer chain polysulfides are produced in subsequent reactions or that they are a minor component of initial reactions. Furthermore, as SOD increased the amount of both per-/polysulfides and thiosulfate produced, (Figure 2 and Figure 3) it appears that superoxide plays little, if any, direct role in oxidation of H_2_S or other sulfur compounds.

Although we could not concomitantly examine the effects of Cat on polysulfide production due to Cat interference with the polysulfide-SSP4 reaction, we were able to examine the effects of both SOD and Cat on thiosulfate production (Figure 4C). Here, it was evident that Cat inhibited spontaneous thiosulfate production from H_2_S, it inhibited thiosulfate production in the H_2_S/1,4-NQ reaction, and it prevented SOD augmentation of thiosulfate production in the H_2_S/1,4-NQ reaction. Collectively, this suggests that hydrogen peroxide derived from superoxide dismutation contributes to some of the end products of H_2_S oxidation catalyzed by 1,4-NQ.

### 3.4. Effects of Ring Substitutions on the H_2_S-NQ Reaction

The two primary factors determining the reactivity of NQs are its capacity to produce free radicals and its electrophilicity [36]. Free radical production by napthoquinones depends primarily on a reduction potential that is high enough to allow efficient reduction by suitable substrates but not so high that it impedes autoxidation, electrophilicity is mainly determined by the electron density at the double bond C2-C3 carbons. Stable hydroquinones hinder electrophilic addition because they are not electrophiles. Ollinger and Brunmark [36] mainly considered free radical production by one-electron reduction of the NQ. However, as we assume reduction by H_2_S is a single two-electron process, free radicals would then be produced by one-electron autoxidation of the hydroquinone. Regardless, they show that the reactions of both 1,4-NQ and juglone are quite favorable, whereas the 2-hydroxyl of lawsone is deprotonated and donates electrons to the quinone ring which hinders reduction of the quinone. Our results affirm that lawsone is not an effective catalyst for H_2_S oxidation,

It is clear from the present experiments that substitutions on the parent 1,4-NQ variously affect H_2_S metabolism. These substitutions could potentially affect either the oxidation of H_2_S or re-oxidation of the NQ. While a detailed examination of these factors was beyond the scope of this study, evidence points to the reduction potentials of H_2_S and the NQs and comproportionation reactions as major contributors.

#### 3.4.1. H_2_S and Reduction of Substituted NQs

The efficacy of the various H_2_S/NQ reactions in terms of oxidation of H_2_S to polysulfides and the attendant oxygen consumption appears to correlate quite well with the two-electron reduction potential of S^0^/HS^−^ (−170 mV) and a two-electron reduction of the various NQs examined herein. These reactions (in decreasing order) and their reduction potentials (in parentheses) are, 1,4-NQ (+83 mV) ≈ juglone (+109 mV) ≈ plumbagin (+20 mV) > 2-methoxy-1,4-NQ (−39 mV) ≈ menadione (−5 mV, or +20 mV) >> phylloquinone (−78 mV) ≈ menaquinone (−70 mV) ≈ lawsone (−140 mV).

#### 3.4.2. Autoxidation of Substituted NQs

Reoxidation (autoxidation) could also be a contributing factor to the reactivity of variously substituted NQs. In fact, this may be the limiting process, but it is clearly more complicated. As described in Section 3.2.4 for 1,4-NQ, autoxidation is unlikely to be a single, two-electron reaction. Of the two, one-electron reactions, oxidation of the hydroquinone to the semiquinone and oxidation of the semiquinone to the quinone, only the second reaction is favorable. Reduction potentials for the second, oxidation of the semiquinone to the quinone, NQ/NQ^·^^−^, are 1,4-NQ (−140 mV), juglone (−93 mV) plumbagin (−156 mV) and menadione (−203 mV). These are very favorable for reactions with superoxide, O_2_^·^^−^/H_2_O_2_, (+910 mV), but not for oxygen O_2_/O_2_^·^^−^ (−180 mV; all oxygen potentials from [42]). This leaves the initial one-electron oxidation of the hydroquinone as the probable limiting process for all the NQs examined in the present study.

Reduced NQs can be autoxidized to the semiquinone by oxygen or semiquinones can result from comproportionation in an oxygen-independent process (Figure 7A–C). The relative importance of each process appears to correlate somewhat with the location and type of ring substitution and the effect of SOD (Figure 7D). Autoxidation is problematic for all NQs because this must start with oxygen, O_2_/O_2_^·−^ (−180 mV), and this does not favor the initial one-electron oxidation of hydroquinones, i.e., NQ^·−^/NQH_2_; 1,4-NQ (+306 mV), juglone (+311 mV), plumbagin (+196 mV) and menadione (+193 mV). SOD provides one way around this as described in Section 3.2.4 by removing one of the reaction products, superoxide. On the other hand, formation of the semiquinone by comproportionation is favored if the NQs readily align. This would be the case for juglone and to a large extent 1,4-NQ. Conversely, we propose that ring substitutions on plumbagin, 2-MNQ and menadione present steric problems for comproportionation. This would lead to an increased dependency on the one-electron autoxidation process which is inherently slow, hence it’s augmentation by SOD.

### 3.5. Reconciliation with Other Studies

Munday [40] examined NQ redox reactions by measuring oxygen consumption when either SOD or oxidized NQs were added to reduced NQs. He observed that the relative rates of autoxidation were, juglone > lawsone > 2-MNQ > menadione > 1,4-NQ, but even 1,4-NQH_2_ was more than 50% oxidized in less than 2 min at pH 7.4. Munday also observed that addition of SOD to reduced 1,4-NQ did not significantly affect the rate of oxygen consumption, whereas SOD increased oxygen consumption by 14% when added to juglone. Conversely, when SOD was added to reduced 2-MNQ, menadione and lawsone, oxygen consumption was inhibited by 83%, 84% and 99% respectively. The inhibitory effect of SOD was attributed to inhibiting semiquinone formation by superoxide (Equation (24)),
NQH_2_ + O_2_^·−^ → NQ^·−^ + H_2_O_2_(24)

When the corresponding oxidized quinone was added to the reduced quinone, oxygen consumption increased by 159%, 34%, 27% and 17% with 1,4-NQH_2_, menadione, 2-MNQH_2_, and lawsone, respectively, but was unaffected with juglone (Table 3). Addition of the oxidized corresponding quinone partially overcame the inhibitory effect of SOD on the autoxidation of 2-MNQ, menadione and lawsone. Munday concluded that, (1) comproportionation was very important in autoxidation of 1,4-naphthohydroquinone, (2) neither comproportionation nor superoxide-driven reactions were significant in the autoxidation of juglone, (3) superoxide was an important chain propagator in the autoxidation of lawsone and comproportionation less important, (4) both chain propagations by superoxide and comproportionation were important in autoxidation of menadione and 2-MNQ.

Ollinger et al. [43] reduced NQs to hydroquionones with DT-diaphorase and observed that SOD inhibited autoxidation of 1,4-NQ and all NQs with a substitution on the quinonid ring, regardless of whether it was -OH, -CH3, or -OCH3. Conversely, SOD stimulated autoxidation when there was an -OH substitution on the benzene ring and this was unaffected by substitutions on the quinoid ring. They concluded that oxygen was important in the inital, slow oxidation and that superoxide generated in this reaction then rapidly oxidized another hydroquinone, which could be inhibited by SOD.

In the present study we used oxidized NQs and reduced them with either H2S, DTT or ascorbic acid (AA) and then examined subsequent reoxidation by measuring oxygen consumption and our results (Table 3) were quite different from those of Munday [40] or Ollinger et al. [43]. When oxidized NQs were initially reduced with H2S, SOD greatly increased O2 consumption by juglone, plumbagin and 1,4-NQ, and to a lesser extent by menadione, 2-MNQ only slightly increased oxygen consumption (although the rate of oxygen consumption appeared to accelerate near the end of the experiment) and lawsone had a negligible effect (Appendix A). By comparison, SOD only modestly increased polysulfide production with H2S and 1,4-NQ, whereas SOD greatly increased polysulfide production H2S and either 2-MNQ or menadione, moderately increased it with plumbagin, but SOD has no effect with juglone or lawsone (Figure 3E). Because SOD did not decrease either polysulfide or thiosulfate production, we assume that the effects of SOD on oxygen consumption were on the reoxidation (autoxidation) of NQs and not on superoxide- or hydrogen peroxide-dependent production of sulfoxides.

Reducing 1,4-NQ with DTT produced a greater increase in oxygen consumption than H_2_S and an even greater increase that consumed all oxygen by 40 min when DTT was added along with H_2_S (Appendix A, respectively). With either DTT + 1,4-NQ or DTT + 1,4-NQ + H_2_S, subsequent addition of SOD initially, and transiently decreased O_2_ consumption. However, DTT did not affect polysulfide production (SSP4 fluorescence) from H_2_S + 1,4-NQ in the absence of SOD, whereas fluorescence was decreased when SOD was added (Appendix A). This suggests that DTT effectively promoted autoxidation of the 1,4-NQ by comproportionation, or that with NQ in the reaction mixture, DTT oxidation competes with sulfide oxidation in the presence of SOD and, since DTT cannot make polysulfides, the polysulfide yield drops. Adding ascorbic acid to reduce 1,4-NQ greatly increased oxygen consumption, either without or with H_2_S, and in both cases SOD had no effect (Appendix A). This is also suggestive of a comproportionation reaction, although we cannot rule out the possibility that superoxide may also be scavenged by DTT and ascorbic acid.

As pointed out by Munday [40] “the mechanism of hydroquinone autoxidation is complex” and by Song and Buettner [12], just because a reaction is “not thermodynamically favorable” does not mean it will not occur. Clearly, other factors such as substrate concentration, pH, solubility, and interactions with non-polar solvents used to dissolve certain NQs, can have substantial effects on these reactions that remain to be fully characterized. Nevertheless, it is interesting that although we did not find evidence to support superoxide as a chain propagator (none of our reactions were inhibited by SOD) and our oxygen consumption results were unlike those of Munday, our conclusions on the relative importance between the initial one-electron autoxidation and comproportionation for the various NQs were generally quite similar.

### 3.6. Significance of Reactive Sulfur Species

Most of the purported health and medicinal benefits of naphthoquinones have been attributed to either pro- or antioxidant effects and these are generally believed to be mediated through modulation of ROS. Here, we provide an alternative mechanism whereby NQs affect metabolism of reactive sulfur species (RSS). The similarities, differences, and advantages of RSS over ROS, especially in terms of biological systems, has been reviewed in detail [44] and are only briefly summarized here. Both oxygen and sulfur are chalcogens with six valence electrons. As sulfur is bigger than oxygen, sulfur’s electrons are farther from the nucleus and being more facile, sulfur will form more varied and stable reaction products. Most biological activity of ROS is attributed to peroxide (H_2_O_2_) which is similar to persulfide (H_2_S_2_), the former the result of a two-electron reduction of molecular oxygen and the latter resulting from two-electron oxidation of 2H_2_S. Both H_2_O_2_ and H_2_S_2_ are oxidants but H_2_S_2_ is also an effective reductant as its pKa is in the physiological range (permitting formation of nucleophilic HS_2_^−^) whereas the pKa for H_2_O_2_ is well above physiological limits. H_2_O is difficult to oxidize and relatively inert from a biological standpoint, whereas H_2_S is not only readily oxidized to other RSS, it also donates electrons to the electron transport chain as a source of energy production and it also contributes to mitochondrial stability. SOD, Cat, and the peroxiredoxin, thiorexodin and glutaredoxin systems, i.e., the endogenous antioxidant defense mechanisms, all metabolize RSS to other biologically active forms of sulfur, but unlike ROS, they do not always irreversibly inactivate RSS. Both H_2_O_2_ and H_2_S_2_ signal via cysteines on regulatory proteins and the effector responses are identical, however, excessive H_2_S_2_ is reversible but excessive H_2_O_2_ is not. Activation of the Keap1-Nrf2 axis by oxidation of Keap1 with H_2_O_2_ (sulfenylation) or persulfidation with H_2_S_2_ is arguably the best example of the duality of these systems and it contributes to the difficulty in resolving the issue of the relative contributions of ROS and RSS in health and disease. We propose that sorting out the relative roles of these two ‘reactive species’ in cellular biology and signaling can help explain the conundrum of many effector pathways activated by NQs. The work presented here illustrates the efficacy of NQ in sulfur metabolism and we suggest that this can explain some, if not much of the therapeutic effects of NQ and related nutraceuticals.

### 3.7. Summary

In summary we have demonstrated that naphthoquinones (NQs) can catalytically produce polysulfides and sulfoxides from hydrogen sulfide by consuming oxygen in a cyclic manner. The polysulfides produced by these reactions are mixtures of different chain lengths of sulfur with the disulfide as the predominate species. NQ-mediated oxidation of H_2_S is increased when SOD is present and decreased when Cat is present implicating superoxide formation in the oxidation of the semiquinone as a rate limiting step, and hydrogen peroxide, derived from superoxide dismutation, as the important compound further oxidizing polysulfides and elemental sulfur to thiosulfate and sulfite. These findings provide the chemical background for a novel sulfur-based approach to naphthoquinone-directed therapies.

## 4. Methods and Materials

### 4.1. H_2_S and Polysulfide Measurements in Buffer

Fluorophore experiments were performed in 96-well plates and fluorescence was measured with a SpectraMax M5e plate reader (Molecular Devices, Sunnyvale, CA. USA). Compounds were pipetted into 96-well plates and the plates were covered with tape to minimize H_2_S loss due to volatilization. Excitation/emission (Ex/Em) wavelengths were per manufacturer’s recommendations; 7-azido-4-methylcoumarin (AzMC, 365/450 nm), 3′,6′-Di(O-thiosalicyl)fluorescein (SSP4, 482/515 nm) and dichlorofluorescein (DCF, 500/525 nm). AzMC and SSP4 have been shown to have sufficient specificity relative to other sulfur compounds and reactive oxygen and nitrogen species (ROS and RNS, respectively) to effectively identify H_2_S (AzMC) and per- and polysulfides (H_2_S_2_ and H_2_S_n_ where *n* = 3–7 or RS_n_H where *n* > 1 or RS_n_R = where *n* > 2) [45,46,47]. As both AzMC and SSP4 are irreversible, they provide a cumulative record of H_2_S and polysulfide production, but they do not reflect cellular concentrations at any specific time.

### 4.2. Oxygen Dependency of Naphthoquinone Reactions with H_2_S in Buffer

H_2_S oxidation by quinones consumes O_2_ [17] and it is likely that NQs do the same. To verify this, O_2_ tension was monitored in a stirred 1 mL water-jacketed chamber at room temperature with a FireStingO_2_ oxygen sensing system (Pyroscience Sensor Technology, Aachen, Germany) using a non-oxygen consuming 3 mm diameter OXROB10 fiberoptic probe. The probe was calibrated with 21% O_2_ (room air) or nitrogen gas (0% O_2_). H_2_S, NQs and other compounds of interest were added at timed intervals and percent O_2_ (100% equals room air) was measured at timed intervals (usually every 0.2–0.3 s) for at least 55 min.

In other experiments the requirement for O_2_ in NQ-catalyzed reactions was examined by conducting parallel experiments in 96 well-plates in the presence (room air) or absence of O_2_, the latter achieved by bubbling phosphate-buffered saline (PBS) with N_2_ gas and conducting the experiments in a model 856-HYPO hypoxia chamber (Plas Labs, Inc. Lansing, MI) set to 0% O_2_ which effectively lowers O_2_ to <1%.

### 4.3. Thiosulfate Production; AgNP Method

Thiosulfate was measured using silver nanoparticles (AgNP) as previously described [48]. Briefly, AgNPs were prepared by reducing AgNO_3_ with tannic acid (TA) in the presence of HAuCl_4_. One mL of 20 mM AgNO_3_ was mixed with 200 μL of 0.5 mM in 98 mL of Milli-Q water at room temperature. One mL of 5.0 mM TA was added, and the mixture was vigorously stirred, turning yellow within 30 min. The AgNPs were stored at 4 °C until use.

For thiosulfate measurements H_2_S and NQs were placed in 96 well plates and the plates were covered with tape for 60 min to minimize H_2_S volatilization during the reaction, the tape was then removed to allow excess H_2_S to dissipate for an additional 60 min. Thirty μL of the buffer was then added to 200 μL of the AgNP in 96-well plates and absorbance measured after 60 min at 419 nm. Thiosulfate standards were made in either buffer or cell medium and the concentrations were plotted against (A_0_ − A)/A_0_ where A_0_ and are the absorbances of AgNPs without and with thiosulfate, respectively. We note that the standard curve is non-linear but reproducible.

### 4.4. Thiosulfate and Sulfite Production; HPLC-MBB Method

Although the AgNP method was useful for most measurements, we noted some interference with other reactions. To confirm our findings, thiosulfate was also measured using high performance liquid chromatography (HPLC, UltiMate 3000, Thermo Fisher Scientific, Waltham, MA, USA) with fluorescence detection (Ultimate FLD-3100 fluorescence detector, Thermo Fisher Scientific) after derivatization with monobromobimane (MBB) as described previously [49,50]. Briefly, 300 μM H_2_S (as Na_2_S) was dissolved in PBS, 10 μM 1,4-NQ was added and allowed to react for 60 min after which it opened for 1 h to remove any residual H_2_S by volatilization. A 50 μL aliquot was then mixed with 5 μL of 25 mM MBB in acetonitrile and incubated in the dark at room temperature for 60 min to convert thiosulfate and sulfite into thiosulfatebimane and sulfitebimane, respectively and subjected to HPLC-fluorescence analysis. A Shim-pack VP-ODS column (5 μm, 4.6 × 150 mm, Shimadzu Scientific, Kyoto, Japan) was used with mobile phases A (MilliQ water containing 0.25% formic acid) and B (methanol containing 0.25% formic acid). Samples (10 μL) were injected onto the column and run with a linear gradient (0–75% B) at a flow rate of 0.8 mL/min and a column temperature of 30 °C. The detector excitation/emission was set to 370/485 nm. Concentrations of thiosulfate and sulfite were determined based on a standard curve prepared from thiosulfate and sulfite standards derivatized with MBB and analyzed on the same day.

### 4.5. Thiosulfate, Sulfite and Sulfate Measurement by Ion Chromatography (IC)

As neither of the above methods could detect sulfate, we employed ion chromatography (IC) to detect sulfate. Although not as sensitive as the above methods, it also provided information on thiosulfate and sulfite.

A Dionex ICS-5000 Ion Chromatography (IC) System (ThermoFisher, Sunnyvale, CA USA) equipped with a conductivity detector, RFIC+ Eluent Degasser, and an AS-AP Autosampler was used to detect the sulfite, sulfate, and thiosulfate. Separation of ions was carried out on a Dionex™ IonPac™ AS14A IC column (4 × 250 mm) attached with a corresponding AS14A guard column (4 × 50 mm). Conductivity was detected with AERS 500 carbonate electrolytically regenerated suppressor. The suppressor voltage, flow rate, and run-time were set at 43 mA, 1.0 mL/min, and 30 min, respectively. Injection volumes on the AS-AP Autosampler were set to 25 μL for analysis. Instrument control and data reduction were carried out with Chromeleon7.2Bsoftware. Data collection rate and temperature were set at 5 Hz and 30 °C, respectively.

Eluent was prepared by diluting AS14A Eluent Concentrate with Milli-Q water to achieve a final concentration of 0.08 mM sodium carbonate/0.01 mM sodium bicarbonate. Sulfite, sulfate, and thiosulfate standard solutions were prepared with Combined Five Anion Standard diluted with Milli-Q water (1:3 *v*/*v*). Sodium thiosulfate (Na_2_S_2_O_3_), sodium sulfite (Na_2_SO_3_), and sodium sulfate (Na_2_SO_4_) standards were dissolved in Milli-Q water and filtered through a 0.22 μm syringe filter (Foxx Life Sciences, Salem, NH, USA) prior to analysis. NQs (300 μM) were added to 3 mM H_2_S and allowed to react in closed Eppendorf tubes for 1.5 h at room temperature. The caps were then opened for another 1.5 h to allow the H_2_S to volatize off and stop the reactions. The samples were filtered through a 0.22 μm syringe filter and collected for IC analysis.

### 4.6. Absorbance Spectra

Absorbance spectra were measured in either 100 or 200 mM PBS, pH 7.4, with either a Shimadzu UV-2401PC recording spectrophotometer (Shimadzu, Kyoto, Japan) with slit width of 0.5 nm, or an Agilent HP 8453 spectrometer (Agilent Technologies, Santa Clara, CA, USA).

In initial experiments, the spectra of oxidized 1,4-NQ (as supplied from the supplier and oxidized by exposure to room air), 1,4-NQ reduced with hydrogen gas (H_2_) and 5% palladium on asbestos or reduced after reaction with H_2_S (as Na_2_S) followed by sparging for 30 min with N_2_ gas to remove the dissolved H_2_S leaving the H_2_S-reduced 1,4-NQ. Quartz cuvettes with screw tops with septa and needles were used for inlet and outlet gassing with N_2_ or room air. In other experiments various ratios of H_2_S and 1,4-NQ were examined in both normoxia (21% O_2_) and hypoxia (~0% O_2_) at 6 s intervals over 1800 s. In these experiments an Na_2_S stock solution prepared in buffer was made anaerobic by N_2_ gas replacement.

### 4.7. Mass Spectrometry

#### 4.7.1. LC-ESI-HRMS Protocol

Inorganic polysulfides were identified using liquid chromatography electrospray ionization high resolution mass spectrometry (LC-ESI-HRMS) using a micrOTOF-Q II Mass Spectrometer (Bruker Daltronics, Billerica, MA, USA) coupled to an UltiMate 3000 (Thermo Fisher) UHPLC system, as described previously [18]. A Waters Acquity UPLC HSS T3 column (1.8 µm, 150 mm × 2.1 mm inner diameter) was used with mobile phases A (water containing 0.1% formic acid) and B (acetonitrile containing 0.1% formic acid). Samples were diluted 10-fold in water and 20 µL was injected with a linear gradient (0–90% B, 30 min) at a flow rate of 0.4 mL/min. The mass spectrometer was used in the positive ion mode with the capillary voltage set to 2200 V and drying gas set to 8.0 L/min at 180 °C. The IAM polysulfide adducts were detected as the [M + Na]^+^ ion using their exact masses ±0.002 *m*/*z*: S_1_ (149.038, 171.020), S_2_ (181.010, 202.992), S_3_ (212.982, 234.964), S_4_ (244.954, 266.936), S_5_ (276.926, 298.908), S_6_ (308.898, 330.880). In these experiments, H_2_S (1 mM, as Na_2_S) was incubated in 10 mM phosphate buffer with 1 mM 1,4-NQ for 10 min, derivatized with 5 mM IAM for 30 min, and subjected to LC-ESI-HRMS within 1–2 h.

#### 4.7.2. Justification for IAM Derivatization

In preliminary studies, H_2_S or K_2_S_x_ (mixture of polysulfides) were derivatized with either iodoacetamide (IAM) or tyrosine methyl ester-iodoacetamide (TME-IAM), the latter having been recently reported to be far superior to IAM and to provide exceptional stability to derivatized polysulfides [51]. IAM or TME-IAM was added to Na_2_S or K_2_S_n_ at 0, 10, 30 and 60 min, analyzed by LC-ESI-HRMS and the area under the curve (AUC) compared. We expected that if derivatization was rapid and the samples were stable then the AUC should be similar, if the derivatization was slower, then the AUC should increase and then plateau. In either case the TME-IAM samples would be superior to the IAM derivatized samples. However, as shown in Appendix A, the AUC for the TME-IAM samples rapidly decreased over time, whereas the IAM derivatized samples did not. We then sparged the buffer with nitrogen to determine if the decrease in AUC in the TME-IAM samples was due to oxidation of polysulfides but this did not appear to be the case (Appendix A). It was evident that IAM derivatized inorganic sulfur samples were considerably more stable than those derivatized with TME-IAM (Appendix A) and all subsequent studies employed IAM derivatization. It should be noted that while the study by Kasamatsu et al. [51] demonstrated the stability of TME-IAM-derivatized organic sulfur compounds, they did not evaluate the stability of inorganic compounds. In studies currently in progress we also found that the TME-IAM was superior for organic polysulfides.

### 4.8. Chemicals

SSP4 was purchased from Dojindo molecular Technologies Inc. (Rockville, MD, USA). All other chemicals were purchased from Sigma-Aldrich (St. Louis, MO, USA) or ThermoFisher Scientific (Grand Island, NY, USA). ‘H_2_S’ is used throughout to denote the total sulfide (sum of H_2_S + HS^−^) derived from Na_2_S as S^2−^ most likely does not exist under these conditions [52]. Phosphate buffered saline (PBS; in mM): 137 NaCl, 2.7 KCl, 8 Na_2_HPO_4_, 2 NaH_2_PO_4_. Phosphate buffer for absorbance measurements (PB; in mM): 200 Na_2_PO_4_. pH was adjusted with 10 mM HCl or NaOH to pH 7.4.

### 4.9. Statistical Analysis

Data was analyzed and graphed using QuatroPro (Corel Corporation, Ottawa, ON, Canada) and SigmaPlot 13.0 (Systat Software, Inc., San Jose, CA, USA). Statistical significance was determined with Students t-test or one-way ANOVA and the Holm-Sidak test for multiple comparisons as appropriate using SigmaStat (Systat Software, San Jose, CA, USA). Results are given as mean + SE; significance was assumed when *p* < 0.05.

## Figures and Tables

**Figure 1 ijms-23-13293-f001:**
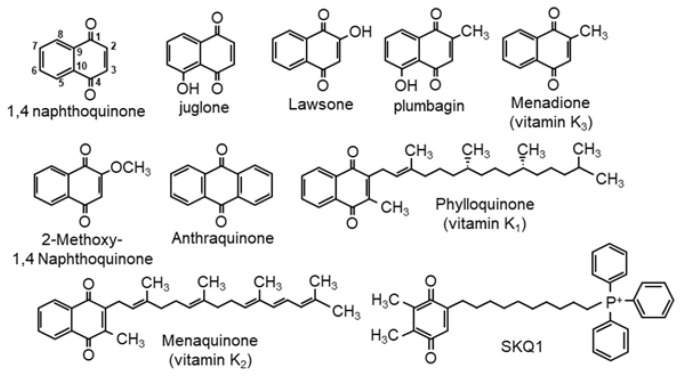
Structures of naphthoquinones used in this study; carbon positions are numbered in 1,4-naphthoquinone for reference.

**Figure 2 ijms-23-13293-f002:**
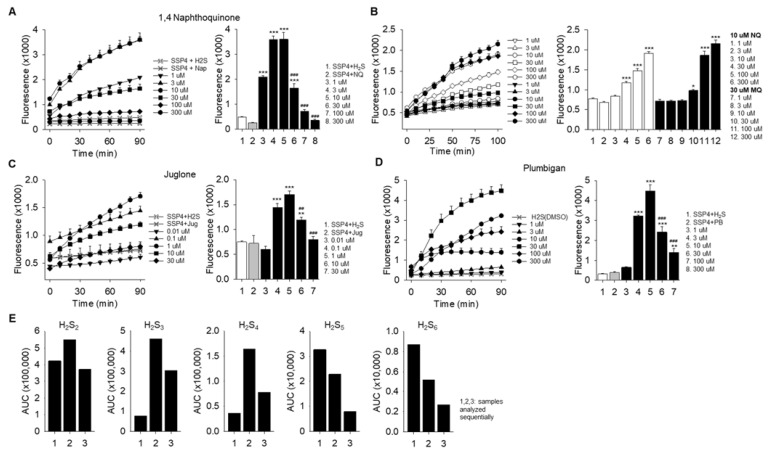
Naphthoquinones oxidize H_2_S to polysulfides (SSP4 fluorescence). (**A**) Effect of increasing concentrations of 1,4-naphthoquinone (1,4-NQ) with constant 300 μM H_2_S, or (**B**) effect of 10 and 30 μM 1,4-NQ with increasing concentrations of H_2_S. (**C**,**D**) Effect of increasing concentrations of juglone (Jug) or plumbagin (Pbn) on SSP4 fluorescence from 300 μM H_2_S. All naphthoquinones initially increased, then decreased SSP4 fluorescence, whereas H_2_S concentration-dependently increased fluorescence at both concentrations of 1,4-NQ. (**E**) LC-ESI-HRMS identification of IAM-derivatized polysulfides produced from 1 mM H_2_S and 1 mM 1,4-NQ. Samples were derivatized after 10 min with 5 mM IAM and three aliquots (1, 2, 3) were analyzed serially at approximately 1 h intervals. (**A**–**D**) Mean + SE, *n* = 4 wells per treatment. *, *p* < 0.05; **, *p* < 0.01; ***, *p* < 0.001 compared to buffer; ##, *p* < 0.01; ###, *p* < 0.001 compared to maximum response.

**Figure 3 ijms-23-13293-f003:**
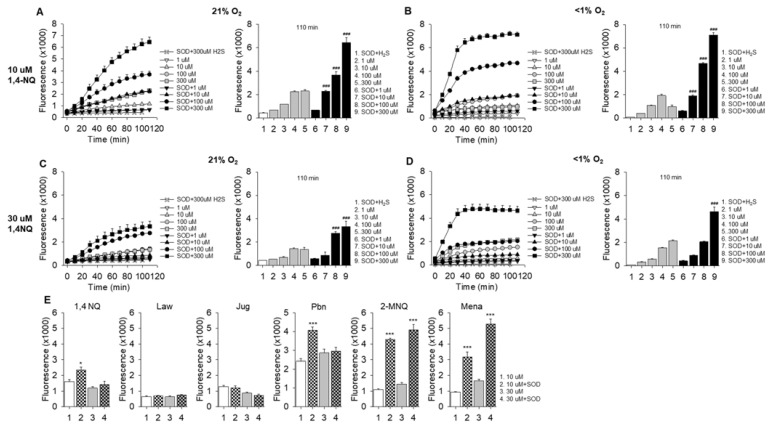
(**A**–**D**) Effects of oxygen (O_2_; 21% and <1%) and superoxide dismutase (SOD; 0.1 μΜ) on polysulfide production (SSP4 fluorescence) by 10 μM or 30 μM 1,4-NQ and 300 μΜ H_2_S. Low O_2_ had minimal effects on polysulfide production, whereas SOD increased production in both 21% and <1% O_2_. Overall polysulfide production was greater with 10 μM than with 30 μM 1,4-NQ. Bar graphs show fluorescence at 110 min. (**E**) Effects of SOD on polysulfide production by 10 μM or 30 μM NQs incubated with 300 μM H_2_S for 120 min. Mean + SE, *n* = 4 wells per treatment; *, *p* < 0.05; ***, *p* < 0.001 compared to similar treatment without SOD. ###, *p* < 0.001 compared to maximum response.

**Figure 4 ijms-23-13293-f004:**
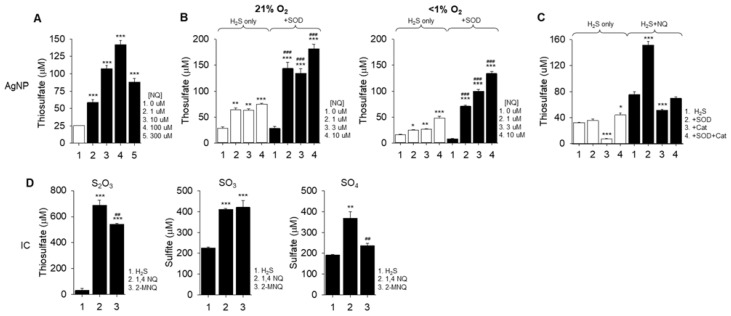
Effects of various parameters on thiosulfate (TS) production from 1,4-NQ (NQ) oxidation of 300 μM H_2_S measured with silver nanoparticles (**A**–**C**) or by ion chromatography (**D**). (**A**) NQ concentration-dependently increased TS production from 1 to 100 μM and decreased it at 300 μM. (**B**) Effects of increasing concentrations of NQ with or without 0.1 μM SOD in TS production in 21% or <1% O_2_. SOD increased TS production at both O_2_ tensions. Low O_2_ somewhat decreased TS production with or without TS but did not inhibit it. (**C**) Effects of 0.1 μM superoxide dismutase (SOD) and 1 μM catalase (Cat), alone or in combination, on TS production from 300 μM H_2_S in the absence (H_2_S only) or presence of 10 μM NQ. With H_2_S alone, TS production was decreased by Cat and slightly increased by SOD + Cat. With H_2_S plus NQ, TS was greatly increased by SOD, decreased by Cat but unaffected by SOD + Cat. (**A**–**C**), Mean + SE, *n* = 8 wells per treatment. *, *p* < 0.05; **, *p* < 0.01; ***, *p* < 0.001 compared to H_2_S from same treatment; (**B**), ###, *p* < 0.001 compared to response without SOD. (**D**) Ion chromatography measurements of production of TS (S_2_O_3_), sulfite (SO_3_) and sulfate (SO_4_) by incubation of 3 mM H_2_S with 300 μM 1,4-NQ or 2-MNQ. Mean + SE, *n* = 3 wells per treatment; **, *p* < 0.01; ***, *p* < 0.001 compared to H_2_S only; ##, *p* < 0.01 compared to 1,4-NQ.

**Figure 5 ijms-23-13293-f005:**
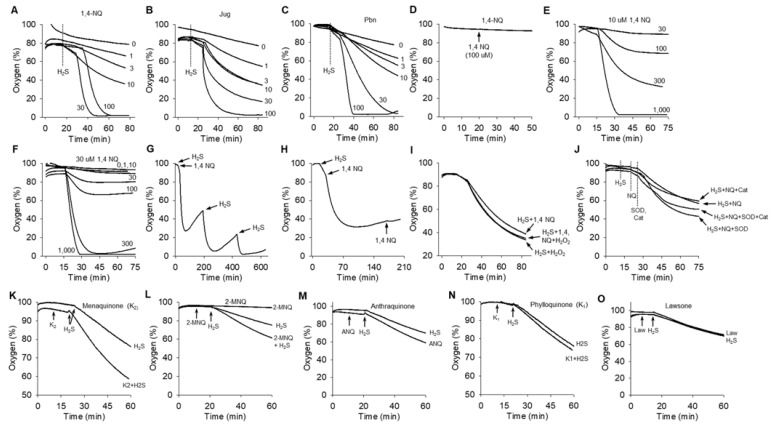
Typical real-time traces of O_2_ consumption by naphthoquinones under various conditions. (**A**–**C**) O_2_ is concentration-dependently consumed during incubation of 300 μM H_2_S with increasing concentrations of (**A**) 1,4-naphthoquinone (1,4-NQ), (**B**) juglone (Jug) or (**C**) plumbagin (Pbn). (**D**) 100 μM 1,4-NQ does not consume O_2_ in the absence of H_2_S. (**E**,**F**) H_2_S concentration-dependently increases O_2_ consumption by 10 μM (**E**) and 30 μM (**F**) 1,4-NQ. (**G**,**H**) O_2_ levels begin to rise after prolonged incubation of 30 μM 1,4-NQ with 300 μM H_2_S and they are decreased upon subsequent addition of H_2_S (**G**) but not by 1,4-NQ (**H**). (**I**) H_2_O_2_ (100 μM) does not affect O_2_ consumption in the presence of 300 μM H_2_S or H_2_S plus 10 μM 1,4-NQ. (**J**) Superoxide dismutase (SOD, 0.1μM) increases O_2_ consumption by 300 μM H_2_S and 10 μM 1,4-NQ, whereas catalase (Cat. 1 μM) decreases it and Cat reduces the effect of SOD. O_2_ consumption by 300 μM H_2_S and 30 μM menaquinone (vitamin K_2_; **K**), 2-methoxy-1,4-NQ (2-MNQ; **L**), anthraquinone (**M**), phylloquinone (vitamin K_1_; **N**) and lawsone (**O**).

**Figure 6 ijms-23-13293-f006:**
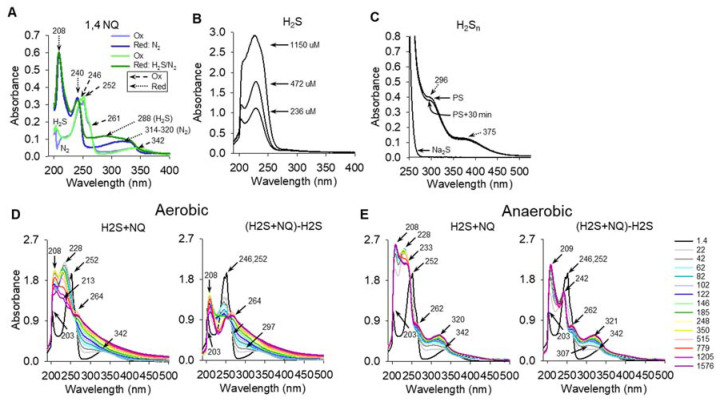
(**A**) Absorbance spectra of oxidized 1,4-NQ (60 μM, light blue/green lines) and after reduction with either N_2_ gas (dark blue line) or 1 mM H_2_S, followed by N_2_ gas to remove excess H_2_S (dark green line). (**B**) Spectra of different concentrations of H_2_S (as Na_2_S); (**C**) spectra of mixed polysulfides (H_2_S_n_) produced by anaerobic incubation of Na_2_S with powdered sulfur (S_8_) in 200 mM phosphate buffer (pH 7.4). Numbers indicate approximate wavelength of peaks. (**D**,**E**) Absorbance spectra as a function of time of the reaction between 230 μM 1,4-NQ and 115 μM H_2_S in aerobic (21% O_2_) and anaerobic (<1% O_2_) buffer. Left panels (H_2_S + NQ) of each pair show the full spectra, in the right panels (H_2_S + NQ) − H_2_S) the H_2_S spectrum was subtracted from the H_2_S + NQ spectrum at each time point. The time, in seconds, after addition of H_2_S is indicated on the right of panel (**E**).

**Figure 7 ijms-23-13293-f007:**
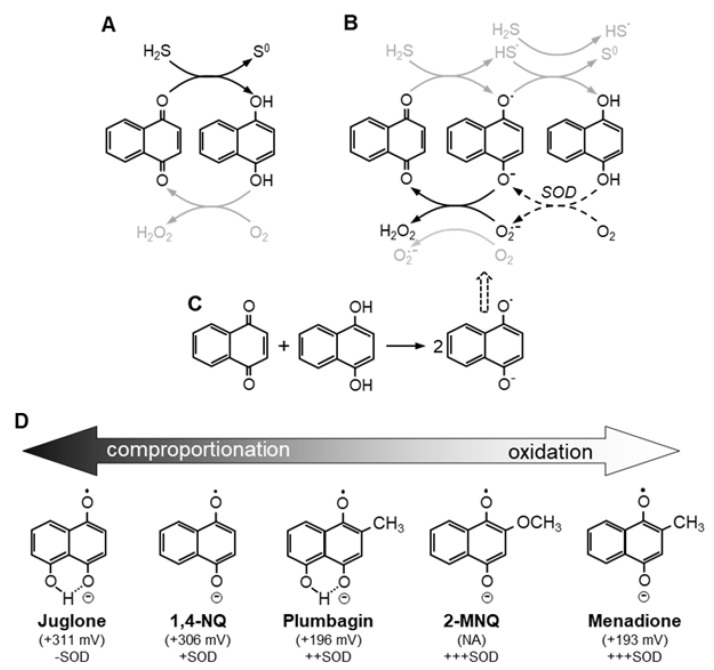
Proposed two-electron (**A**) and one-electron (**B**) reactions of 1,4-NQ with H_2_S and (**C**) the comproportionation reaction generating the semiquinone from oxidized and reduced 1,4-NQ. Gray arrows and chemical pathways indicate less favorable reactions, dashed gray arrows show pathway enhanced by SOD-catalyzed removal of superoxide. (**D**) Potential effects of ring substitutions on autoxidation or comproportionation in the initial one-electron oxidation of naphthohydroquinones. Naphthoquinones that readily comproportionate are minimally affected by SOD. Reduction potentials in in parentheses, -SOD, minimal effect of superoxide dismutase (SOD). +SOD, +++SOD moderate and large effect of SOD, respectively.

**Table 1 ijms-23-13293-t001:** Rate constants A → B (*k*_1_) and B → C (*k*_2_) for deconvoluted UV-Vis absorbance spectra of the H_2_S reaction with 1.4 naphthoquinone (NQ).

μM NQ:μM H_2_S	*k* _1_	*k* _2_
115:236	6.7 × 10^−3^ s^−1^	1.5 × 10^−3^ s^−1^
230:236	17.0 × 10^−3^ s^−1^	2.0 × 10^−3^ s^−1^
230:472	21.0 × 10^−3^ s^−1^	7.8 × 10^−3^ s^−1^
230:1150	22.0 × 10^−3^ s^−1^	3.4 × 10^−3^ s^−1^

**Table 2 ijms-23-13293-t002:** Reduction potentials and relative toxicity (LC_50_) of naphthoquinones used in this study.

NQ	pKa NQ^·−^, QH_2_	Q/NQ^·−^	NQ^·−^/QH_2_	Q/QH_2_	LC_50_ (μM) ^1^	Superoxide Formation ^1^
1,4-NQ ^(2)^	4.1, 9.4	−140 ^a^	+306	+83	15	↑
Juglone ^(2,3,5)^	6.6	−93, −95	+311	+109, +82	6	↑
Plumbagin ^(3)^		−156	+196	+20	5	↑↑
Lawsone ^(2,5)^	4.7, 8.7	−415 ^b^	+135	−140, −86	>100	→/↑
Vit K_1 (4)_ (phylloquinone)				−78		
Vit K_2_ (menaquinone)				−70		
Vit K_3_ (menadione) ^(3,5)^	9.6	−203, −335	+193	−5, +20	40	↑↑↑
Benzoquinone ^(3)^		+99	+473	+286		
2-methoxy 1,4-NQ ^(3)^	9.4			−39		
Anthraquinone		−445				

^1^ Ref. [11], ^2^ Ref. [36] and references therein, ^3^ Ref. [12] and references therein, ^4^ Ref. [39], ^5^ Ref. [40] and references therein as half-wave potentials relative to NHE, pKa from [36,40]. Up arrows indicate degree of increase, horizontal arrow indicates no change. a, [40]; b, addition of corresponding oxidized NQ.

**Table 3 ijms-23-13293-t003:** Comparison of the relative rates of oxygen consumption (O_2_) during autoxidation of NQs under various conditions from Munday ^a^ and this study. Right column shows polysulfide production (SSP4 fluorescence) by SOD with NQ and H_2_S. a, [40]; b, addition of corresponding oxidized NQ; c, [18]. Up arrows indicate degree of increase, horizontal arrow indicates no change and down arrows indicate degree of decrease.

	Munday ^a^		This Study				
	SOD (O_2_)	NQ_(ox)_ (O_2_) ^b^	H_2_S (O_2_)	H_2_S + SOD (O_2_)	DTT (O_2_)	DTT + SOD (O_2_)	H_2_S + SOD (SSP4)
1,4-NQ	**→**	**↑↑↑**	**↑↑↑**	**↑↑**	**↑↑↑**	**↓↑**	**↑**
2-MNQ	**↓↓↓**	**↑↑**	**↑**	**↑**	**→**	**↑**	**↑↑↑**
Juglone	**↑**	**→**	**↑↑↑**	**↑↑↑**	**↑↑↑**	**↑**(slight)	**→**
Menadione	**↓↓↓**	**↑↑**	**↑**	**↑**	**↑↑**	**↓**	**↑↑↑** ^c^
Lawsone	**↓↓↓**	**↑**	**↑**	**→**(**↑↓**)	**→**	**→**	**→**

## Data Availability

Data is contained within the article or Appendix A.

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
