# Peer review of "Naphthoquinones Oxidize H2S to Polysulfides and Thiosulfate, Implications for Therapeutic Applications"

_ijms, 2022, doi:10.3390/ijms232113293_

Round 1
Reviewer 1 Report
To Authors:
In this manuscript, the authors studied Reactive sulfur species (RSS) generation by 1,4-Napththoquinones (NQs) oxidation of hydrogen sulfide (H2S) using several methods to provide a chemical background for novel sulfur-based approaches to naphthoquinone-directed therapies.
Herein, the authors showed that NQs oxidize H2S to polysulfides using RSS-specific fluorophores (SSP4) or mass spectrometry (Fig. 2). low O2 had minimal effects on polysulfide production, whereas superoxide dismutase (SOD) increased production in both 21% and <1% O2 (Fig. 3). Using silver nanoparticles and ion chromatography, NQs were also shown to oxidize HS to thiosulfate and sulfite (Fig. 4). Furthermore, they showed that O2 was consumed in the reaction of NQs with H2S (Fig. 5) and that the absorbance spectra were consistent with the NQs redox cycle (Fig. 6).
While the results support the authors' conclusion, the following minor concerns should be addressed before publication in the journal.
Major points:
1. The authors used several NQs in this paper, and, interestingly, differences in polysulfide production and oxygen consumption are observed due to differences in their structures. The paper would be better if more descriptions of structure-activity relationships were added.
2. Polysulfides are inherently reactive and thus unstable. Although IAM and MBB are used in this paper, speciation of complex mixtures of reactive sulfur species using their relatively strong electrophiles is very difficult. However, recent reports indicate that the IAM derivatives HPE-IAM and TME-IAM stabilize them. TME-IAM detected longer polysulfide than IAM in the results shown in the supplementary material of this paper. The reason the results seem more stable with IAM is likely due to the accumulation of shorter sulfides as a result of the decomposition of polysulfides. Therefore, if authors want to be more precise about the number of sulfur atoms in polysulfides, authors should use a more stable probe such as TME-IAM instead of IAM or MBB. Also, since the TME-IAM is used without such explanation, the relevant paper should be cited.
3. Regarding fig. 2E, does this mean that the times after the IAM reaction are different because Aliquots 1, 2, and 3 were measured in that order? The time, temperature, and pH at that time may affect the stability of polysulfide, so it is better to indicate the conditions. Also, showing the results without NQ addition makes it easier to compare because the amount of polysulfide carried in from the original Na2S becomes clear.
4. The authors showed that oxygen consumption by NQs and H2S was unaffected by H2O2 addition, but did not show the effects of H2O2 addition on polysulfide and thiosulfate production. If data are available, showing them will allow us to better understand the effect of SOD on this reaction.
Miner points:
1. In Fig. 2B, unlike the others, the fluorescence intensity increased with 30 µM NQ compared to 10 µM. Could this also be due to the variation in SSP4 measurements mentioned in the text?
2. In lane 121 of p3, isn't H2S3 a mistake of H2S2?
3. In lane 124 of p3, sample 1 does not match the percentage of total polysulfide detected, so isn't it a mistake in sample 2? Or please rewrite the number of percentages.
4. In Fig. 6, the description of PS is unclear but is it powdered sulfur?
5. Please write the legend of the number in the right figure of Fig. 2C as well as others.
6. In fig. 7, isn't it "one electron (A) and two electrons (B) reactions of 1,4-NQ with H2S" mistake of "two electrons (A) and one electron (B) reactions"?
7. In Supplementary Figure S1, it is described as K2S (K2Sn), does that mean it is a mixture? Also, please describe how to adjust K2Sn.
Author Response
Reviewer 1
To Authors:
In this manuscript, the authors studied Reactive sulfur species (RSS) generation by 1,4-Napththoquinones (NQs) oxidation of hydrogen sulfide (H2S) using several methods to provide a chemical background for novel sulfur-based approaches to naphthoquinone-directed therapies.
Herein, the authors showed that NQs oxidize H2S to polysulfides using RSS-specific fluorophores (SSP4) or mass spectrometry (Fig. 2). low O2 had minimal effects on polysulfide production, whereas superoxide dismutase (SOD) increased production in both 21% and <1% O2 (Fig. 3). Using silver nanoparticles and ion chromatography, NQs were also shown to oxidize HS to thiosulfate and sulfite (Fig. 4). Furthermore, they showed that O2 was consumed in the reaction of NQs with H2S (Fig. 5) and that the absorbance spectra were consistent with the NQs redox cycle (Fig. 6).
While the results support the authors' conclusion, the following minor concerns should be addressed before publication in the journal.
Major points:
- The authors used several NQs in this paper, and, interestingly, differences in polysulfide production and oxygen consumption are observed due to differences in their structures. The paper would be better if more descriptions of structure-activity relationships were added.
* We have extensively rewritten section 3.4. to address this among with an additional study on the effects of SOD on H2S oxidation by plumbagin (Fig. 3E). This is now summarized in Fig. 7D. We believe this helps, although these reactions are admittedly complex.
- Polysulfides are inherently reactive and thus unstable. Although IAM and MBB are used in this paper, speciation of complex mixtures of reactive sulfur species using their relatively strong electrophiles is very difficult. However, recent reports indicate that the IAM derivatives HPE-IAM and TME-IAM stabilize them. TME-IAM detected longer polysulfide than IAM in the results shown in the supplementary material of this paper. The reason the results seem more stable with IAM is likely due to the accumulation of shorter sulfides as a result of the decomposition of polysulfides. Therefore, if authors want to be more precise about the number of sulfur atoms in polysulfides, authors should use a more stable probe such as TME-IAM instead of IAM or MBB. Also, since the TME-IAM is used without such explanation, the relevant paper should be cited.
*We appreciate the reviewer’s points regarding the stability of polysulfides (PS) and the inherent problems associated with their measurement. Although IAM has been used for some time to derivatize PS, we assumed the TME-IAM method would be preferable based on the recent report by Kasamatsu et al., 2021 (now cited). However, as we reoprted, this did not appear to be the case as the IAM samples appeared to be more stable. It should be noted that Kasmatsu et al. did not examine the stability of inorganic polysulfides, and in studies in progress in our lab we do confirm that the TME-IAM method is superior for organic sulfur compounds. To clarify the issue we have added the following subsection:
“4.7.2. Justification for IAM derivatization
In preliminary studies, H2S or K2Sx (mixture of polysulfides) were derivatized with either iodoacetamide (IAM) or tyrosine methyl ester-iodoacetamide (TME-IAM), the latter having been recently reported to be far superior to IAM and to provide exceptional stability to derivatized polysulfides (Kasamatsu et al., 2021{13017}). IAM or TME-IAM was added to Na2S or K2Sn at 0, 10, 30 and 60 min, analyzed by LC-ESI-HRMS and the area under the curve (AUC) compared. We expected that if derivatization was rapid and the samples were stable then the AUC should be similar, if the derivatization was slower then the AUC should increase and then plateau. In either case the TME-IAM samples would be superior to the IAM derivatized samples. However, as shown in Supplemental Fig. S1A, B, the AUC for the TME-IAM samples rapidly decreased over time, whereas the IAM derivatized samples did not. We then sparged the buffer with nitrogen to determine if the decrease in AUC in the TME-IAM samples was due to oxidation of polysulfides but this did not appear to be the case (Supplemental Fig. S1C). It was evident that IAM derivatized inorganic sulfur samples were considerably more stable than those derivatized with TME-IAM (Supplemental Figure S1) and all subsequent studies employed IAM derivatization. It should be noted that while the study by Kasamatsu et al. (Kasamatsu et al., 2021{13017}) demonstrated the stability of TME-IAM-derivatized organic sulfur compounds, they did not evaluate the stability of inorganic compounds. In studies currently in progress we also found that the TME-IAM was superior for organic polysulfides.” We hope this is sufficient.
- Regarding fig. 2E, does this mean that the times after the IAM reaction are different because Aliquots 1, 2, and 3 were measured in that order? The time, temperature, and pH at that time may affect the stability of polysulfide, so it is better to indicate the conditions. Also, showing the results without NQ addition makes it easier to compare because the amount of polysulfide carried in from the original Na2S becomes clear.
* We agree that this is not clear. This sentence has been changed to:
“IAM (10 mM) was then added and the sample was divided into three aliquots (Samples 1, 2, 3) which were serially analyzed at approximately 1 h intervals to monitor both the identity and stability of the IAM-polysulfides.”
The stability of IAM derivatized H2S and H2S2 samples without NQ were described in the Supplement (see above). We hope this is sufficient because of the high cost of repeating the LC-ESI-HRMS experiments.
- The authors showed that oxygen consumption by NQs and H2S was unaffected by H2O2 addition, but did not show the effects of H2O2 addition on polysulfide and thiosulfate production. If data are available, showing them will allow us to better understand the effect of SOD on this reaction.
* We have shown that H2O2 only slowly (if at all in these conditions) oxidizes H2S to polysulfides. We added the following “This suggests that hydrogen peroxide contributes to much of the thiosulfate produced in these reactions, although it may not substantially contribute to polysulfude production from H2S (ref is Olson et al., 2017 Redox Biol 15:74:85).” We hope this sufficient.
Miner points:
- In Fig. 2B, unlike the others, the fluorescence intensity increased with 30 µM NQ compared to 10 µM. Could this also be due to the variation in SSP4 measurements mentioned in the text?
* Yes, we think that the reviewer is correct. We did not feel that an in depth examination of this would be necessary as it is clear from the figures that the SSP4 is further depressed at higher [NQ] and the H2S effect is clearly concentration-dependent in both cases.
- In lane 121 of p3, isn't H2S3 a mistake of H2S2?
* Correct. This has been changed, thank you!
- In lane 124 of p3, sample 1 does not match the percentage of total polysulfide detected, so isn't it a mistake in sample 2? Or please rewrite the number of percentages.
* We thank the reviewer for catching this! Our error was in inserting the wrong H2S2 panel in figure 2E, the percentages are correct. We have inserted the correct figure A, also included percentages for the second sample in the text and a hypothesis of the reactions.
“As shown in Fig. 2E, multiple inorganic polysulfides were produced after 10 min incubation of H2S with 1,4-NQ. The area under the curve (AUC) for H2S2, H2S3 and H2S4 increased between samples 1 and 2 but decreased for H2S5 and H2S6. The AUC for all samples decreased from sample 2 to sample 3. This suggests that either polysulfides, or IAM-derivatized polysulfides with n>4, are somewhat unstable and may initially decompose to lower molecular weight polysulfides over the first hour and then later decompose to other sulfur moieties that are undetectable by LC-ESI-HRMS. Using values from sample 1, the percentage of the total polysulfide detected at 10 min was 39, 42, 15, 3 and 0.8% for S2-6, respectively, and for sample 2 was 57, 32, 8 2 and 1%. Collectively, this suggest that a variety of polysulfides are produced by 1,4-NQ catalyzed oxidation of H2S and that H2S2 and possibly H2S3 are likely the primary products of this reaction.”
We believe that this data further supports our conclusion that S2 and S3 are the predominant polysulfides produced in these reactions. We hope this clarifies the confusion.
- In Fig. 6, the description of PS is unclear but is it powdered sulfur?
* Yes, the mixed polysulfides were produced by mixing powdered sulfur with H2S (Na2S).
- Please write the legend of the number in the right figure of Fig. 2C as well as others.
* Done, not sure how this got deleted, our apologies.
- In fig. 7, isn't it "one electron (A) and two electrons (B) reactions of 1,4-NQ with H2S" mistake of "two electrons (A) and one electron (B) reactions"?
* Yes you are correct and changed accordingly, thank you!
- In Supplementary Figure S1, it is described as K2S (K2Sn), does that mean it is a mixture? Also, please describe how to adjust K2Sn.
*K2S has been changed to K2Sn. This salt produces a variety of polysulfides as well as H2S when dissolved. Here we show the characterization and distribution of these different compounds. As we point out the actual characterization of these compounds may vary with the time and method used.
Reviewer 2 Report
This article describes the chemical reaction of 1,4-naphthoquinone (1,4-NQ) and derivatives with H2S. Under the conditions of changing the concentrations of different substrates, the reaction products were measured in the ways of fluorescent reagents, ion chromatography, mass spectrometry, and absorption spectrometry. The consumption of oxygen was also measured. It was confirmed that NQs and H2S first completed the reduction of NQs and the oxidation of H2S by two-electron transfer. After that, it is likely that NQH2 completed the first-step one-electron reaction through two mechanisms to form a semiquinone intermediate (NQ·-) and superoxide (O2·-). Then, the oxidized NQs are formed by the second step one-electron oxidation with concomitant formation of hydrogen peroxide. It is speculated that hydrogen peroxide will further oxidize sulfane sulfur to form other sulfuroxides.
While the authors argue this reaction in detailed and offered a lot of evidences, I still have questions about some of the details and think the article can be further optimized:
Major concern:
1. The article is not easy to read, and there are many repetitive descriptions. English can also be improved.
2. Fig. 4B:The saturated dissolved oxygen in water is 8 mg/L. Under nonoxic condition (<1% O2), the oxygen concentration in buffer is generally significantly lower than this concentration. It is unclear how to form high S2O32- content under this condition. Where does the oxygen atom in thiosulfate come from?
3. Fig. S1: The authors believe that the it is more stable to use IAM to alkylate H2Sn, not TME-IAM. Puzzlingly, why the authors can detect so high intensity of H2Sn signal at 0 hr? According to the author's experimental design, IAM and TME-IAM should not react with H2S or H2Sn at 0 hr.
4. In the discussion section, the content in 3.4.2 and 3.2.4 are partly overlapping. It is better to combine them.
Minor concern:
1. Line 159: “We have previously shown that low oxygen decreases” The sentence is confusing.
2. Line 160: “CoQ0” , or “CoQ”?
3. Fig. 3: um?
4. Line 194: In Fig. S4, most of the fluorescence intensity did not reach highest in 20 min.
5. Fig. 4B: It is “thiosulfate”.
6. Line 264-265:“These results confirm the AgNP 264 and HPLC-MBB studies and add sulfate to the list of species produced by NQ oxidation 265 of H2S.” The sentence is confusing.
7. Line 311-312: “SOD did not 311 produce a further increase in oxygen consumption but appeared to produce an initial 312 transient decrease the rate of oxygen consumption (Supplemental Fig. S6A).” The sentence is confusing.
8. Line 507:“(eq. 2; here indicated……” Missing closing bracesï¼›
9. Line 604: What is meaning of “SQ”?
Author Response
Reviewer 2:
This article describes the chemical reaction of 1,4-naphthoquinone (1,4-NQ) and derivatives with H2S. Under the conditions of changing the concentrations of different substrates, the reaction products were measured in the ways of fluorescent reagents, ion chromatography, mass spectrometry, and absorption spectrometry. The consumption of oxygen was also measured. It was confirmed that NQs and H2S first completed the reduction of NQs and the oxidation of H2S by two-electron transfer. After that, it is likely that NQH2 completed the first-step one-electron reaction through two mechanisms to form a semiquinone intermediate (NQ·-) and superoxide (O2·-). Then, the oxidized NQs are formed by the second step one-electron oxidation with concomitant formation of hydrogen peroxide. It is speculated that hydrogen peroxide will further oxidize sulfane sulfur to form other sulfuroxides.
While the authors argue this reaction in detailed and offered a lot of evidences, I still have questions about some of the details and think the article can be further optimized:
Major concern:
- The article is not easy to read, and there are many repetitive descriptions. English can also be improved.
* We have revised the ms to correct these issues.
- Fig. 4B The saturated dissolved oxygen in water is 8 mg/L. Under nonoxic condition (<1% O2), the oxygen concentration in buffer is generally significantly lower than this concentration. It is unclear how to form high S2O32- content under this condition. Where does the oxygen atom in thiosulfate come from?
* Thank you for this comment. We do not know but we suspect some diffusion of O2 into the reaction mixture. Estimates of O2 range from 180 to 265 uM in 300 mosm buffer which as the reviewer points out would imply 1-2 uM O2 in deoxygenated solutions. Unfortunately, we did not have the resources to measure thiosulfate by other methods. Clearly this needs further investgation but we feel that the main point of this was to show that thiosulfate was also produced from NQ oxidation of H2S.
- Fig. S1: The authors believe that the it is more stable to use IAM to alkylate H2Sn, not TME-IAM. Puzzlingly, why the authors can detect so high intensity of H2Sn signal at 0 hr? According to the author's experimental design, IAM and TME-IAM should not react with H2S or H2Sn at 0 hr.
* H2S and polysulfides are produced immediately upon dissolution of Na2S and K2Sn so it is not surprising to see them react with IAM or TME-IAM, essentially as soon as the alkaylating agents are added to these solutions.
- In the discussion section, the content in 3.4.2 and 3.2.4 are partly overlapping. It is better to combine them.
* We prefer to keep them separate as section 3.2. deals mainly with the mechanisms involved with 1,4-NQ, whereas 3.4. discussed the vagaries of ring substitutions. We feel that combining them would be disrupt the flow of each section. However, both sections have been revised to better focus on their respective points.
Minor concern
- Line 159 “We have previously shown that low oxygen decreases” The sentence is confusing.
* This sentence has been changed to “We have previously shown that low oxygen decreases SSP4-detected polysulfide production when H2S is incubated with CoQ0, whereas in normoxic solutions SOD increases polysulfides when H2S is incubated with CoQ0...” We hope this is satisfactory and apologize for the confusion.
- Line 160: “CoQ0” , or “CoQ”
* CoQ0 is correct.
- Fig. 3: um?
* added.
- Line 194: In Fig. S4, most of the fluorescence intensity did not reach highest in 20 min.
* We understand, but this was a choice that was made by the corresponding author (KRO) to allow the experiments to be completed in one day, mostly by undergraduates with restricted schedules. It was felt that their inclusion in these experiments was an important part of their education and it was also our opinion that this duration was sufficient to discern the important differences.
- Fig. 4B: It is “thiosulfate”.
* Corrected.
- Line 264-265 “These results confirm the AgNP 264 and HPLC-MBB studies and add sulfate to the list of species produced by NQ oxidation 265 of H2S.” The sentence is confusing.
* This has been rewordrd to “These results support the observations that thiosulfate was detected by both AgNP and HPLC-MBB methods and they add sulfate to the list of species produced by NQ oxidation of H2S.” We hope this clears it up.
- Line 311-312: “SOD did not 311 produce a further increase in oxygen consumption but appeared to produce an initial 312 transient decrease the rate of oxygen consumption (Supplemental Fig. S6A).” The sentence is confusing.
* This has been changed to “SOD did not augment oxygen consumption by DTT and 1,4-NQ but, instead appeared to produce an initial transient decrease the rate of oxygen consumption” In addition the arrow and ‘SOD’ have been moved to the other side of the trace to better illustrate which trace experienced the SOD treatment. We hope this clarifies the issue.
- Line 507 “(eq. 2; here indicated……” Missing closing braces
* Corrected, thank you.
- Line 604: What is meaning of “SQ”?
* This has been changed to “(NQ/SQ; where SQ is the semiquinone)’ We hope this is sufficient.